# Prevalent and dynamic binding of the cell cycle checkpoint kinase Rad53 to gene promoters

Yi-Jun Sheu, Risa Karakida Kawaguchi[†], Jesse Gillis[‡], Bruce Stillman*

Cold Spring Harbor Laboratory, Cold Spring Harbor, United States

**Abstract** Replication of the genome must be coordinated with gene transcription and cellular metabolism, especially following replication stress in the presence of limiting deoxyribonucleotides. The *Saccharomyces cerevisiae* Rad53 (CHEK2 in mammals) checkpoint kinase plays a major role in cellular responses to DNA replication stress. Cell cycle regulated, genome-wide binding of Rad53 to chromatin was examined. Under replication stress, the kinase bound to sites of active DNA replication initiation and fork progression, but unexpectedly to the promoters of about 20% of genes encoding proteins involved in multiple cellular functions. Rad53 promoter binding correlated with changes in expression of a subset of genes. Rad53 promoter binding to certain genes was influenced by sequence-specific transcription factors and less by checkpoint signaling. However, in checkpoint mutants, untimely activation of late-replicating origins reduces the transcription of nearby genes, with concomitant localization of Rad53 to their gene bodies. We suggest that the Rad53 checkpoint kinase coordinates genome-wide replication and transcription under replication stress conditions.

*For correspondence:
stillman@cshl.edu

Present address: [†]Center for iPS Cell Research and Application, Kyoto University, Kyoto, Japan; [‡]Department of Physiology, Donnelly Centre, University of Toronto, Toronto, Canada

## Editor's evaluation

The unexpected localization of a cell cycle checkpoint kinase, Rad53, to promoters in response to replication stress suggests that Rad53 may help coordinate transcription in response to disrupted replication. This work will be of interest to those interested in the interplay between genome stability and gene expression.

## Introduction

Rad53, the homolog of the mammalian CHEK2 kinase present in the budding yeast *Saccharomyces cerevisiae*, is an essential kinase involved in a multitude of cellular processes. The most well-established function of Rad53 is its role as an effector kinase in response to various sources of DNA damage. To maintain genome stability during S-phase, a DNA replication checkpoint (DRC) is activated in response to replication stress via the sensor kinase Mec1 (the mammalian ATM/ATR), the replication fork protein Mrc1 (Claspin in mammals), and other fork proteins (*Lanz et al., 2019*; *Osborn and Elledge, 2003*; *Pardo et al., 2017*; *Paulovich and Hartwell, 1995*; *Saldivar et al., 2017*). A second DNA damage checkpoint mediated by Rad9 (TP53BP1 in mammals) responds to double strand DNA breaks. Both branches converge on the effector kinase Rad53, which triggers a wide range of downstream events, including slowing or halting cell cycle progression, transiently preventing initiation of DNA replication at origins that have not yet replicated, activating DNA repair pathways and elevating the synthesis of deoxyribonucleoside triphosphates (dNTPs) by upregulating genes encoding the ribonucleotide reductase (RNR). The checkpoint signaling also promotes widespread changes in gene expression (*Jaehnig et al., 2013*; *Pardo et al., 2017*).

Unlike most of the checkpoint genes, both Mec1 and Rad53 kinases are essential for cell viability in unperturbed cells. This can be partly explained by their role in regulating dNTP pools (*Desany et al., 1998*; *Forey et al., 2020*; *Zhao et al., 2000*). However, under the bypass conditions in cells without Sml1, the inhibitor of the RNR enzyme, cells lacking either kinases are viable. Yet, it is important to note that these kinase-null mutants are extremely sick and sensitive to various types of exogenous stress. Furthermore, lacking Rad53 causes a more severe defect than lacking Mec1, implying that Rad53 has activities beyond checkpoint signaling. Consistent with this suggestion, the kinase-deficient mutant *rad53*[K227A] lacks checkpoint function but retains growth-associated activity and an additional *rad53* mutation was found that is checkpoint proficient but supports cell growth poorly (*Gunjan and Verreault, 2003*; *Hoch et al., 2013*; *Holzen and Sclafani, 2010*; *Pellicioli et al., 1999*).

Rad53 is central to the transcriptional response to DNA damage. It is well established that the Dun1 protein kinase acts downstream of Rad53 to phosphorylate and inactivate the transcriptional repressor Rfx1/Crt1 and thereby upregulate target genes (*Huang et al., 1998*), including *RNR2*, *RNR3*, and *RNR4*, encoding subunits of RNR. However, the induced expression of *RNR1*, which encodes the major isoform of the RNR large subunit, is not controlled by the Rfx1 repressor, but by Ixr1 binding to the *RNR1* promoter upon genotoxic stress (*Tsaponina et al., 2011*). This Ixr1-dependent regulation of *RNR1* is independent of Dun1 but requires Rad53. Another Rad53-dependent, Dun1-independent regulation of *RNR1* involves dissociation of the Nrm1 repressor from MBF following Nrm1 phosphorylation by Rad53 (*Travesa et al., 2012*). Thus, the Rad53-dependent transcription control is complex, but not yet fully understood.

In addition to upregulating the dNTP pools, defects in cells lacking Rad53 can be suppressed by manipulating factors functioning in transcription regulation, cell wall maintenance, proteolysis, and cell cycle control (*Desany et al., 1998*; *Manfrini et al., 2012*). Moreover, Rad53 kinase targets and interaction partners found in biochemical and proteomic studies suggest that the kinase is pleiotropic (*Gunjan and Verreault, 2003*; *Jaehnig et al., 2013*; *Lao et al., 2018*; *Smolka et al., 2007*; *Smolka et al., 2006*).

In this study, while investigating the response of yeast cells to replication stresses caused by depletion of dNTPs, we found that Rad53 localized to more than 20% of gene promoters in the *S. cerevisiae* genome, suggesting a multifaceted role in coordinating stress responses. Furthermore, we provide evidence that in the absence of the DRC checkpoint pathway, untimely activation of replication from late origins can negatively affect transcription activity of nearby genes.

## Results

### Rad53 is recruited to genomic loci other than replication origins in proliferating yeast cells

Previous studies focusing on specific genomic features, mainly subsets of DNA replication origins, have detected Rad53 binding to replication origins and replication forks (*Can et al., 2019*; *Dohrmann and Sclafani, 2006*). However, given the evidence that Rad53 has functions beyond the control of DNA replication stress, it is possible that Rad53 also functions at additional genomic loci. We therefore investigated the genome-wide distribution of Rad53 by chromatin immunoprecipitation and deep sequencing (ChIP-seq) in proliferating yeast cells (*Behrouzi et al., 2016*). Since the *sml1* null mutation (*sml1Δ*) allows cells to bypass the requirement for Rad53, or its upstream kinase Mec1, for growth, we also performed ChIP-seq in the *sml1Δ* mutant and the *rad53* null (*rad53Δ sml1Δ*) as controls for antibody specificity (*Figure 1*).

Visual inspection of the ChIP-seq peaks from normalized coverage tracks suggested that Rad53 bound to sites throughout the genome (*Figure 1a*). Rad53 binding to the same sites also occurred in the *sml1Δ* mutant and the *mec1Δ sml1Δ* mutant, but was absent in the *rad53Δ sml1Δ* mutant. Computational analysis was performed to generate heatmaps of Rad53 ChIP-seq signal across 2-kb intervals centered on transcription start sites (TSSs) (*Figure 1b*). The average Rad53 ChIP-seq signal on top of the heatmap show that Rad53 was most concentrated within 500-bp upstream of TSSs in *WT*, *sml1Δ*, and *mec1Δ sml1Δ* mutants. A smaller fraction of genes also have signal within the gene bodies. No Rad53 signal beyond background was detected at TSSs in the *rad53Δ sml1Δ* cells, demonstrating antibody specificity. Analysis by calculating empirical p values for the ChIP-seq signals 500-bp upstream of TSS found that, out of 6604 genes, 1464, 1983, and 1293 in *WT*, *sml1Δ*, and *mec1Δ*

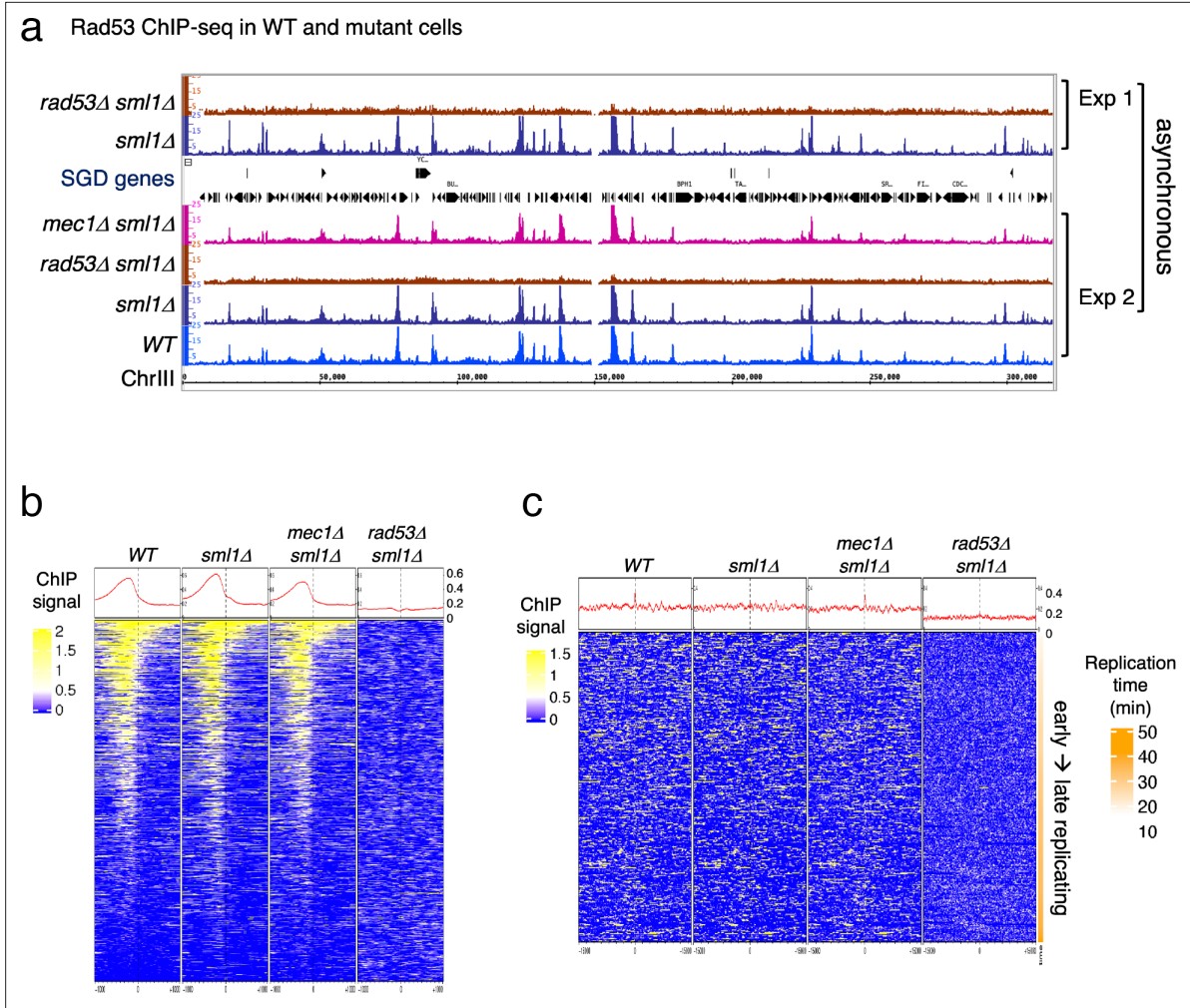

**Figure 1.** Rad53 is recruited to genomic loci other than replication origins in proliferating yeast cells. (**a**) Coverage tracks of Rad53 ChIP-seq signals in *WT*, *sml1Δ*, *rad53Δsml1Δ*, and *mec1Δ sml1Δ* for chromosome III. Asynchronous yeast cultures were processed for ChIP-seq analysis for distribution of Rad53 at genomic loci. The results from two independent experiments are presented. Experiment 1 compares only *sml1Δ*, *rad53Δsml1Δ*. (**b**) Heatmaps and average signals of Rad53 across 2-kb intervals centered on transcription start sites (TSSs) for proliferating *WT*, *sml1Δ*, *rad53Δsml1Δ*, and *mec1Δ sml1Δ* cells. (**c**) Heatmaps of ChIP-seq signals across 30-kb intervals centered on all origins annotated in OriDB database (*Siow et al., 2012*).

*sml1Δ*, respectively, exhibited significant Rad53 signal at this region. This represents ~20% or greater of all gene promoters.

Heatmap analysis was also performed on Rad53 ChIP-seq signal across 30-kb intervals centered on all origins annotated in the OriDB database (*Siow et al., 2012*). The average signal across the whole region was higher in *WT*, *sml1Δ*, and *mec1Δ sml1Δ* than in *rad53Δ sml1Δ* after normalization (*Figure 1c*). However, the signal was not concentrated at the origins. Since these ChIP-seq datasets are from asynchronous cells, this finding is consistent with the idea that recruitment of Rad53 to replication origins is cell cycle regulated rather than constant binding to origins.

## Binding of Rad53 to upstream TSS changes with cell cycle stages

To gain insight into the dynamics of Rad53 recruitment to genomic loci such as promoters and replication origins, Rad53 ChIP-seq was analyzed in samples from synchronized cell cultures. Cells were arrested in G1 using α-factor and then released into media containing hydroxyurea (HU) to induce replication stress caused by limiting dNTPs. Three stages of synchronous cell cultures were collected and referred to as G1 (for cells arrested in G1), HU45 and HU90 (for cells released from G1 into HU for 45 and 90 min, respectively). These cell samples were then processed for ChIP-seq analysis. Heatmaps of the Rad53 signals at 2-kb intervals centered on all TSSs show a trend of increasing Rad53 binding

as cells progress from G1-phase into HU45 or HU90 (*Figure 2a*), concomitant with increased levels of Rad53 protein in cells treated with HU (*Figure 2—figure supplement 1*). The increase in Rad53 parallels entry into S-phase, as measured by Orc6 phosphorylation (*Figure 2—figure supplement 1a*). Similar to the data from asynchronous cell samples, Rad53 signals were most concentrated upstream of TSSs (*Figure 2a*, left panel). The ChIP-seq signals using antibodies against γ-H2A, which like Rad53 is a target of the sensor kinase Mec1, were not enriched upstream of TSSs (*Figure 2a*, right panel). In fact, the signals for γ-H2A were lower immediately upstream of TSSs than the surrounding regions, consistent with promoters being histone-free regions.

Further analysis showed that >85% of the Rad53 ChIP-seq peaks overlap with gene promoters (defined as 500-bp upstream and 50-bp downstream of TSS) at all stages investigated. The distribution of aggregated peak numbers around TSSs showed highest count numbers immediately upstream of the TSS (*Figure 2b*). By visual inspection of normalized ChIP-seq coverage tracks, we noticed that signals at promoters of genes such as *RNR1*, *PCL1*, and *TOS6* varied depending on the cell cycle stage (*Figure 2c*, bottom three tracks for *WT* at stages of G1, HU45, and HU90) while signals at neighboring gene promoters remained largely constant. Thus, the recruitment of Rad53 to a subset of gene promoters is regulated. Because *RNR1*, *PCL1*, and *TOS6* are known targets of the cell cycle regulator SBF, a sequence-specific transcription factor composed of subunits Swi4 and Swi6, we compared Rad53 ChIP-seq data and a previous ChIP-seq dataset of Swi6 (*Park et al., 2013*). As indicators for the gene specificity of protein binding, the Gini indices were computed from Lorenz curves of ChIP-seq data for Swi6 and two of our Rad53 replicates (*Figure 2d*), being 0.763, 0.2918, and 0.2982, respectively. Rad53 has a coverage for many promoters while Swi6 shows substantially high coverage only for a limited number of promoters. Thus, it is likely Rad53 would effect a wider range of genes than the Swi6 regulatory network.

Previous studies have found that under certain conditions, regions of the genome are promiscuously present in ChIP-seq studies independent of the antibody used (*Park et al., 2013*; *Teytelman et al., 2013*). These regions were termed hyper-ChIPable regions and were found enriched for sequences in and around gene bodies of highly expressed genes. We therefore examined whether these regions were promiscuously present under our conditions. Analysis of ChIP-seq data for Rad53 and γ-H2A with or without the hyper-ChIPable regions observed by Teytelman et al. and Park et al. did not alter the pattern of Rad53 binding to TSSs (*Figure 2—figure supplement 2a–d*). When we specifically examined the pattern of Rad53 ChIP-seq signals around TSSs of the 296 genes associated with those hyper-ChIPable sequences, we observed not only localization to TSSs, but also enrichment to the gene bodies as previously reported (*Park et al., 2013*; *Teytelman et al., 2013*). This pattern is distinct from the pattern of promoter localization (compare *Figure 2—figure supplement 2d, e*). Furthermore, KEGG analysis of the genes enriched in the studies by Teytelman et al. and Park et al. showed predominantly genes encoding snoRNAs and tRNAs, genes we did not find in the promoter binding for Rad53 (*Figure 2—figure supplement 2f*). Finally, we did not observe any enrichment at gene promoters when the *RAD53* gene was deleted from the strain (*Figure 1a, b*) or when anti-γ-H2A antibodies were used (*Figure 2a*). Thus, we suggest that the Rad53 binding observed here is not the same as the promiscuous, non-specific enrichment of hyper-ChIPable regions reported previously. Moreover, Rad53 binding to a subset of promoters is transcription factor dependent (see below).

## Rad53 is recruited to sites of DNA synthesis independent of checkpoint signaling

Since previous studies reported localization of Rad53 to replication origins (*Can et al., 2019*; *Dohrmann and Sclafani, 2006*), we also performed heatmap analysis of ChIP-seq signal around replication origins for Rad53, γ-H2A, and Cdc45, a component of active helicase complex and hence the marker for active replication forks (*Figure 2—figure supplement 3*). The 30-kb window for the heatmap around origins was chosen based on our prior knowledge that the extent of DNA synthesis in *WT* cells under similar growth conditions at HU90 was about 10 kb (*Sheu et al., 2014*). In *WT* cells, the Rad53 signal was present at regions associated with early firing origins but not with late firing origins that are inactive due to inhibition by the HU-induced checkpoint. However, Rad53 signal was also present at late origins in the kinase-deficient *rad53^{K227A}* and *mrc1Δ* mutants, both of which allow activation of late origins as a result of the checkpoint defect. Thus, Rad53 was recruited only to replication forks associated with activated origins and the pattern was similar to that of Cdc45. Surprisingly, Rad53 binding

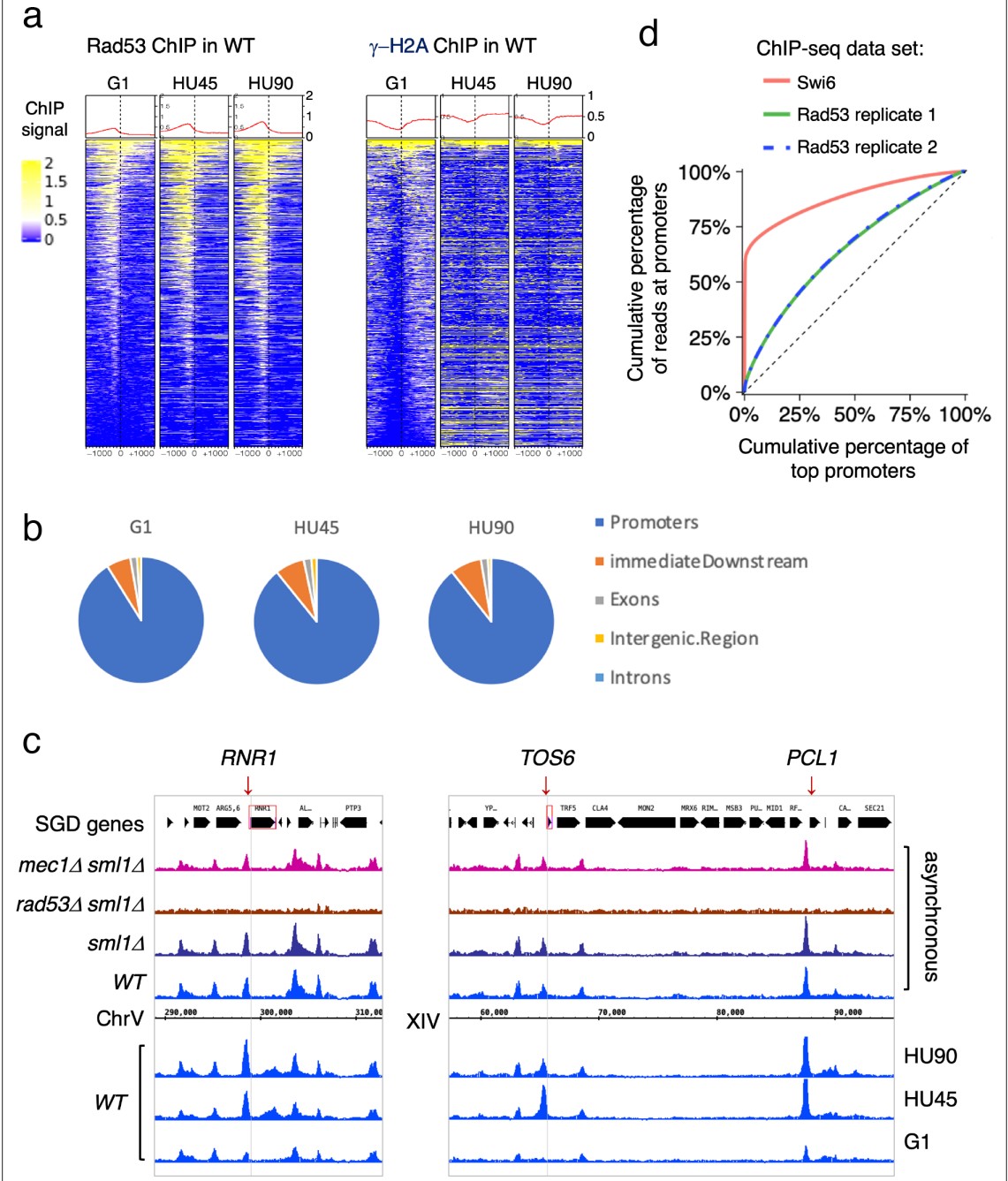

**Figure 2.** Rad53 is recruited to transcription start site (TSS) and the binding changes with the cell cycle stage. Cells were synchronized in G1 phase and released into YPD containing 0.2M hydroxyurea (HU) for 45 and 90min (HU45 and HU90, respectively). (**a**) Heatmaps and average signals of Rad53 and γ-H2A ChIP-seq signals across 2-kb intervals centered on TSSs for *WT* cells at stages of G1, HU45, and HU90. (**b**) Distribution of aggregated peak numbers around TSS using merged Rad53 ChIP-seq peaks from all three stages (G1, HU45, and HU90). Pie charts showing the distribution of Rad53 ChIP-seq peaks in relation to gene features. (**c**) Rad53 ChIP-seq profiles near *RNR1*, *PCL1*, and *TOS6* genes in proliferating *WT*, *sml1Δ*, *rad53Δ sml1Δ*, and *mec1Δ sml1Δ* cells, and *WT* cells at stages of G1, HU45, and HU90. (**d**) Lorenz curves for ChIP-seq read counts of Swi6 and Rad53 mapped to gene promoters showing inequality for promoter binding.

The online version of this article includes the following source data and figure supplement(s) for figure 2:

**Figure supplement 1.** Relative level of Rad53 protein changes in cells.

**Figure supplement 1—source data 1.** Rad53 and Orc6 in *Figure 2—figure supplement 1a*.

**Figure supplement 1—source data 2.** γ-H2A in *Figure 2—figure supplement 1a*.

*Figure 2 continued on next page*

*Figure 2 continued*

**Figure supplement 1—source data 3.** Sml1 in *Figure 2—figure supplement 1a*.

**Figure supplement 1—source data 4.** Rad53 in *Figure 2—figure supplement 1b*.

**Figure supplement 1—source data 5.** Orc6 in *Figure 2—figure supplement 1b*.

**Figure supplement 1—source data 6.** γ-H2A and Sml1 in *Figure 2—figure supplement 1b*.

**Figure supplement 1—source data 7.** Rad53 in *Figure 2—figure supplement 1c*.

**Figure supplement 1—source data 8.** Orc6 in *Figure 2—figure supplement 1c*.

**Figure supplement 1—source data 9.** γ-H2A in *Figure 2—figure supplement 1c*.

**Figure supplement 1—source data 10.** All images used in *Figure 2—figure supplement 1a*.

**Figure supplement 1—source data 11.** All images used in *Figure 2—figure supplement 1b*.

**Figure supplement 1—source data 12.** All images used in *Figure 2—figure supplement 1c*.

**Figure supplement 2.** Comparison of ChIP-seq signals of γ-H2A and Rad53 with and without sequences found enriched promiscuously in previous studies.

**Figure supplement 3.** Heatmaps of ChIP-seq signal across 30kb centered on all active origins.

**Figure supplement 4.** Rad53 ChIP-seq profiles near *RNR1*, *PCL1*, and *TOS6* genes in at stages of G1, HU45, and HU90.

to replication forks did not require Mrc1 or its own kinase activity, suggesting checkpoint-independent recruitment of Rad53 to sites of DNA synthesis.

Interestingly, γ-H2A was observed at genomic regions surrounding the very late origins in G1-phase in both *WT* and mutants (*Figure 2—figure supplement 3b*). It is possible that these γ-H2A signals reflect a low level of ssDNA gaps at these late-replicating regions that were tolerated and carried over from the previous cell cycle, similar to unrepaired post-replication gaps resulting from low level of UV irradiation found in *S. pombe* G2-phase (*Callegari and Kelly, 2006*).

## Identification of genes with differential binding of Rad53 at promoters

Rad53 promoter binding was temporally dynamic in a subset of genes, suggesting regulation by cell cycle progression or DNA replication stress. To identify genes with differential or dynamic binding of Rad53 at their promoters (DB genes), we applied residual analysis. The read count difference was investigated for the promoter regions (500-bp window upstream of TSSs) of all genes for Rad53 ChIP-seq. The comparison was done between stages G1 and HU45 of *WT* samples from two independent experiments (termed CP and TF, *Figure 3a*). Since each dataset had two biological replicates from stages G1 and HU45, we extracted the top 1000 genes that displayed dynamic binding from the aggregated read coverage and called these genes the top DB genes (*Figure 3a*). Among the 1000 top DB genes from each set of comparison, 435 genes were identified in both sets (435 top DB overlap).

Overall, during the G1- to S-phase transition (HU45), there are more genes with increased Rad53 promoter binding than those with decreased binding: within the list of 435 genes, 337 show increased Rad53 binding at their promoters, while 98 show decreased binding (*Figure 3b*). These genes include those involved in cell cycle progression (e.g., genes encoding cyclins and regulators of DNA replication) and cell growth (e.g., cell wall maintenance and mating response). *Figure 3c* shows Rad53 dynamic binding, either up or down, at the promoters of representative genes as cells transitioned from G1 phase to HU45 and HU90 time points.

## The relationship between Rad53 promoter binding and gene expression

To gain insight into the relationship between Rad53 promoter recruitment and gene expression, RNA-seq analysis was performed under the same growth conditions used for the ChIP-seq experiments. RNA-seq replicates of datasets from four yeast strains (*WT*, *rad9Δ*, *rad53^{K227A}*, and *mrc1Δ*), each with three stages (G1, HU45, and HU90) were analyzed using rank data analysis (*Figure 4a*). In the hierarchical clustering, cell cycle stage contributes more to similarities in expression than the genotype. In particular, the expression profiles in G1 were very similar among all strains. In HU45, however, two subgroups clustered by genotype were evident: *rad9Δ* clustered with *WT*, consistent with Rad9 having no role in the DRC checkpoint branch, while *rad53^{K227A}* and *mrc1Δ* clustered together, consistent with Rad53 and Mrc1 functioning together in the DRC response to HU stress.

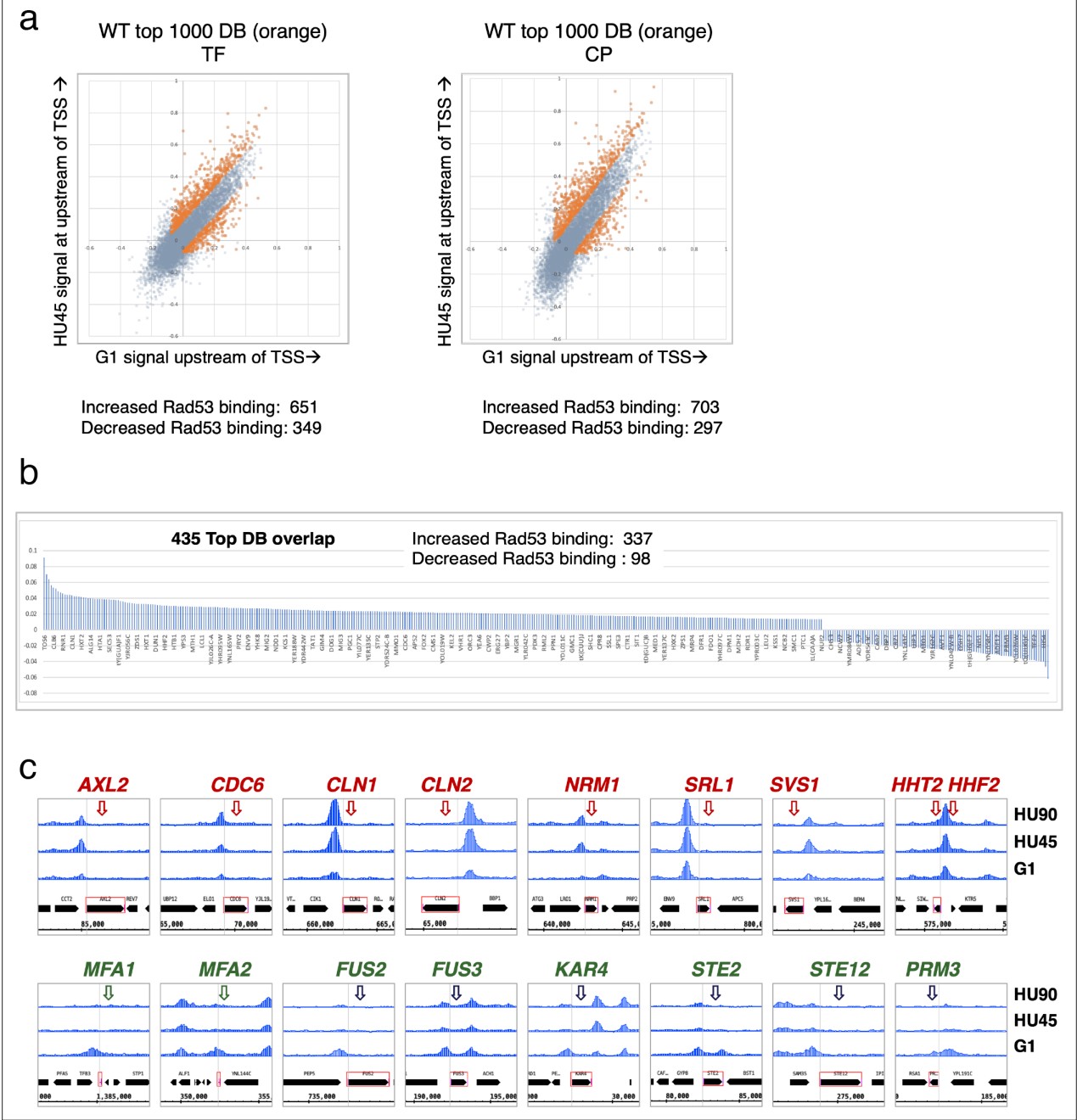

**Figure 3.** Identification of genes with Rad53-binding changes at the promoters. (**a**) Scatter plots compare the signals in G1 (*x*-axis) and HU45 (*y*-axis) at 500-bp intervals upstream of transcription start sites (TSSs) for all genes in *WT* datasets from two independent experiments, TF and CP. Orange dots indicated the 1000 genes with highest binding changes (top 1000 DB) and satisfying the filter of minimal signal of −0.075 (Maximal = 1). The Venn diagram to the right illustrates the identification of 435 genes that are found in both experiments (435 top DB overlap). (**b**) Binding changes (*y*-axis: residuals from analysis of experiment TF) for 435 top DB overlap. (**c**) Examples of coverage tracks for selected genes show Rad53 signal changes at the indicated gene promoters from G1 to HU45 and HU90.

Pairwise comparisons of G1 to HU45 samples showed that, in both *WT* and *rad9Δ* cells, ~2300 genes exhibited significant expression changes (differentially expressed genes, DEGs; *Figure 4b*). The number of DEGs increased to ~3000 when comparing G1 to HU90. Moreover, in both *rad53*$^{K227A}$ and *mrc1Δ* mutants, ~2500 DEGs were detected from G1 to HU45, which increased to >3400 in G1 to HU90. The response to cell cycle stage was largely equally distributed between up- and downregulation. Comparison of *WT* to *rad53*$^{K227A}$ in the HU45 and HU90 conditions found 517 and 2234 DEGs,

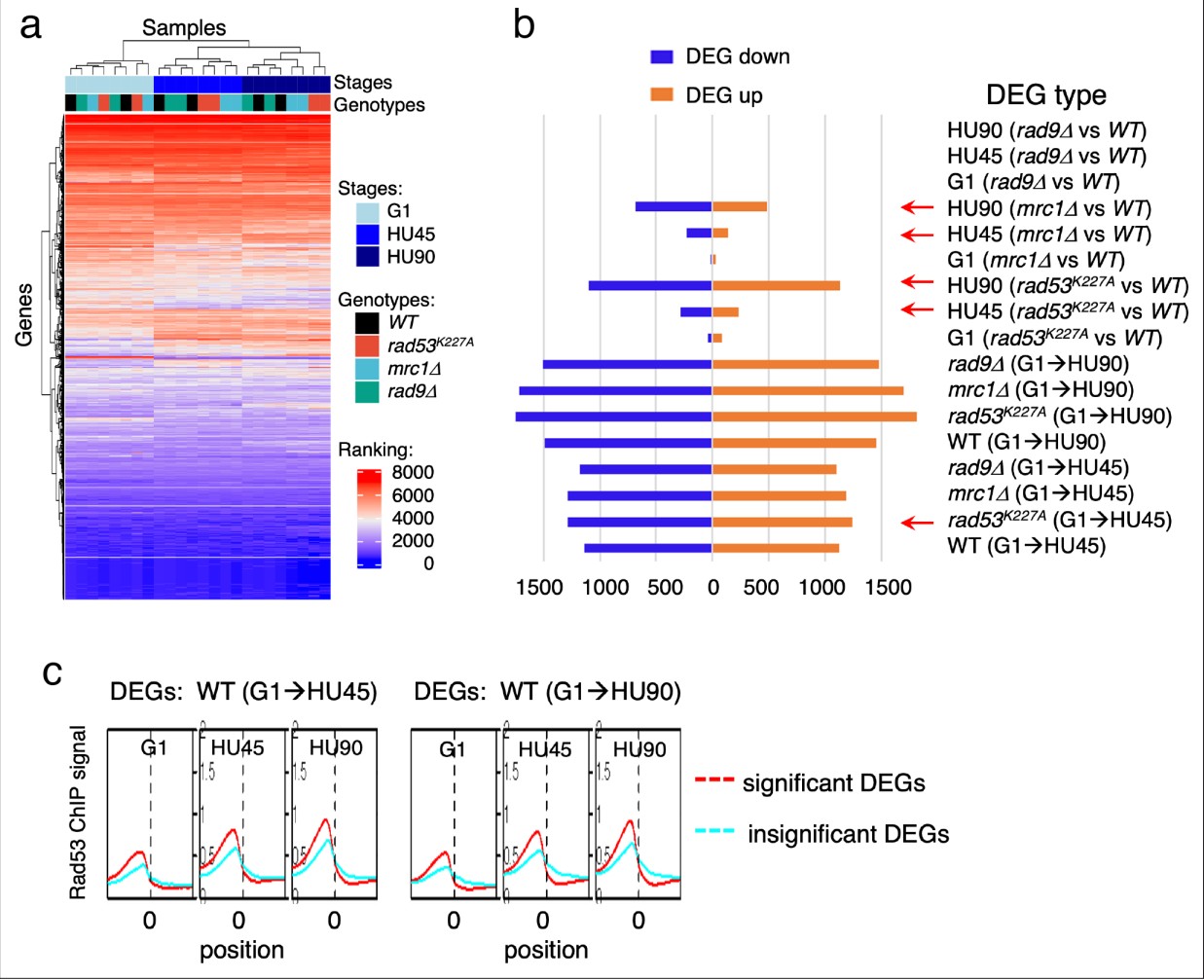

**Figure 4.** Gene expression changes in *WT* and checkpoint mutants under stress and the tendency of higher Rad53 binding at promoter of genes with significant differential expression. *WT*, *rad9Δ*, *rad53*^K227A^, and *mrc1Δ* cells were synchronized in G1-phase and released into YPD containing 0.2M hydroxyurea (HU). Cells at stages of G1, HU45, and HU90 were collected and processed for RNA-seq analysis. (**a**) Rank data analysis of RNA-seq samples. (**b**) Bar graph summarizing on the *x*-axis the number of genes that show statistically significant differential expression (DEGs). The types of pairwise comparison are indicated to the right. Blue bars, downregulated DEGs. Orange bars, upregulated DEGs. (**c**) Average Rad53 ChIP-seq signal across 2-kb intervals centered on at transcription start site (TSS) for statistically significant DEGs (red) and the insignificant (cyan). Left panel shows the result of DEGs from comparison of G1 and HU45 datasets of *WT* (DEGs: *WT*(G1 → HU45)) and right panel shows DEGs from comparison of G1 and HU90 datasets (DEGs: *WT*(G1 → HU90)).

The online version of this article includes the following figure supplement(s) for figure 4:

**Figure supplement 1.** Average Rad53 ChIP-seq signal and heatmaps of signal across 2-kb intervals centered on transcription start site (TSS) for stage-dependent differentially expressed genes (DEGs).

respectively, while comparison of *WT* to *mrc1Δ* revealed 370 and 1166 DEGs in HU45 and HU90. A *WT* and *rad9Δ* comparison at all stages showed only five DEGs, including the deleted *RAD9* gene and its marker cassette *HIS3*. Thus, Rad9 did not contribute much to gene expression changes under HU stress.

Based on ChIP-seq data, the average Rad53-binding upstream of TSSs was higher in the significant DEGs than in the insignificant, when comparing G1 to HU45 or to HU90 of the *WT* datasets (*Figure 4c* and *Figure 4—figure supplement 1a*). These results suggested that Rad53 may play a role in the control of gene expression at a subset of expressed genes. Similar observations were also apparent when comparing stage-dependent samples in the checkpoint mutants (*Figure 4—figure supplement 1b, c*), demonstrating that the functions of Rad53 at gene promoters were not solely checkpoint dependent. Therefore, we performed gene coexpression analysis based on association in the

coexpression network (*Lee et al., 2020*). Coexpression analysis of significant DEGs in *WT*(G1 → HU45) yielded 10 coexpression clusters (*Figure 5a*). Dynamic Rad53 binding at promoter regions occurred in most clusters (*Figure 5b*), with significantly enriched GO functions including cell cycle regulation, mating response, proteolysis, transport, oxidation–reduction process, and organic acid metabolism (*Figure 5a*). Within the 435 top DB overlap gene group (*Figure 3b*), 236 genes were also detected as DEGs (236 DB/DEGs) in the *WT*(G1 → HU45) comparison. The association between binding changes and expression changes was significant (*Figure 5d*, left panel; Fisher's exact test p < 0.0001). Plots of Rad53-binding changes against gene expression changes of these 236 DB/DEGs revealed a moderate positive correlation between Rad53-binding change and gene expression change, with a Spearman *r* = 0.5918 (*Figure 5c*, left panel). Among this group, 51 out of 54 genes with decreased Rad53 signal were downregulated in mRNA levels, whereas genes with increased Rad53 signals partitioned into both up- and downregulation (108 and 74, respectively). Further break down of the 236 DB/DEGs group by coexpression clusters revealed that genes in clusters 1 and 7 exhibited the most significant positive correlation between Rad53-binding and gene expression changes (*Figure 5c*, *r* = 0.5697 (p < 0.000001) and *r* = 0.6979 (p < 0.000037), respectively). Thus, specific subsets of DEGs in the shift from G1 → HU exhibited correlations between a change in gene expression and Rad53 promoter binding.

## Checkpoint mutants cause downregulation of gene expression near promiscuously active late origins

We further inspected the localization of Rad53 in a subset of DEG clusters from the HU45 (*mrc1Δ* vs. *WT*) comparison (*Figure 6a*) and the HU45 (*rad53^{K227A}* vs. *WT*) comparison (*Figure 6—figure supplement 1a*). In these analyses, we noticed a characteristic pattern, in which downregulated genes tended to have a strong Rad53 signal not only upstream of the TSS, but a broad signal within gene bodies (*Figure 6b* and *Figure 6—figure supplement 1b*). This pattern was prominent in the *mrc1Δ* mutant in HU45 and further intensified in HU90. The gene body localization was also found more transiently in HU45 sample from *rad53^{K227A}* cells. This signal pattern was not as prevalent in the *WT* HU45 and HU90 samples. Since Rad53 is also recruited to active origins and moves with the replication fork, we suspected that these gene body signals in the checkpoint mutants were caused by the promiscuous activation of near-by origins that are normally inactive in *WT*, creating conflicts between DNA replication and gene transcription. The transient nature of the Rad53 localization at gene bodies in this group of genes in the *rad53^{K227A}* mutant is also consistent with the transient signal pattern at origins (*Figure 2—figure supplement 3c*). Thus, we investigated the relationship between these genes and their closest replication origins.

The distance from the TSS of each gene to the nearest replication origins, the relative orientation of gene transcription to the origin (head-on [HO] or co-directional [CD]) and the origin type (early, late or inactive; see Definition of the origin types in Materials and methods) was determined and presented in the same order as in the heatmaps of DEG clusters shown in *Figure 6b, c* (*Figure 6c*). Overall, most of the downregulated genes in cluster 1 of this group were situated very close to active origins (<2 kb between origin center and TSS, light purple marks and <1 kb, dark purple marks). Interestingly, the pattern of marks for origin to TSS distance largely mirrored the patten of the Rad53 ChIP signal within the gene bodies (*Figure 6b, c*). This correlation pattern was not found in the *WT* ChIP heatmaps.

To explore the functional relationship between replication origins and genes, we summarized the ratio of up- and downregulated DEGs from the HU45 (*mrc1Δ* vs. *WT*) comparison in terms of the category of their relation to the closest origins (*Figure 6d*). In the first category, we parsed this DEG group according to the range of distance between TSS and closest origins (*Figure 6d*, left panels). Within this DEG group, genes situated 5 kb or more away from the closest active origins were similarly distributed between up- and downregulation of gene expression. However, for those genes that are closer to an active origin (<5 kb), the bias toward downregulated genes increased. For those genes situated less than 1 kb away from active origins, more than 80% were downregulated. In the second category, we grouped the DEGs according to the type origin the TSS was closest to. The DEGs that are more than 5 kb away from any annotated origins were labeled 'none' in origin type (*Figure 6d*, middle panels) and similarly distributed between up- and downregulation as expected from the above analysis. However, more downregulated genes were found when the nearby origins were active (*Figure 6d*, origin type late or early). The bias was stronger for genes that were close to late origins, which become active in HU when Mrc1 was absent (86% and 71% downregulated when

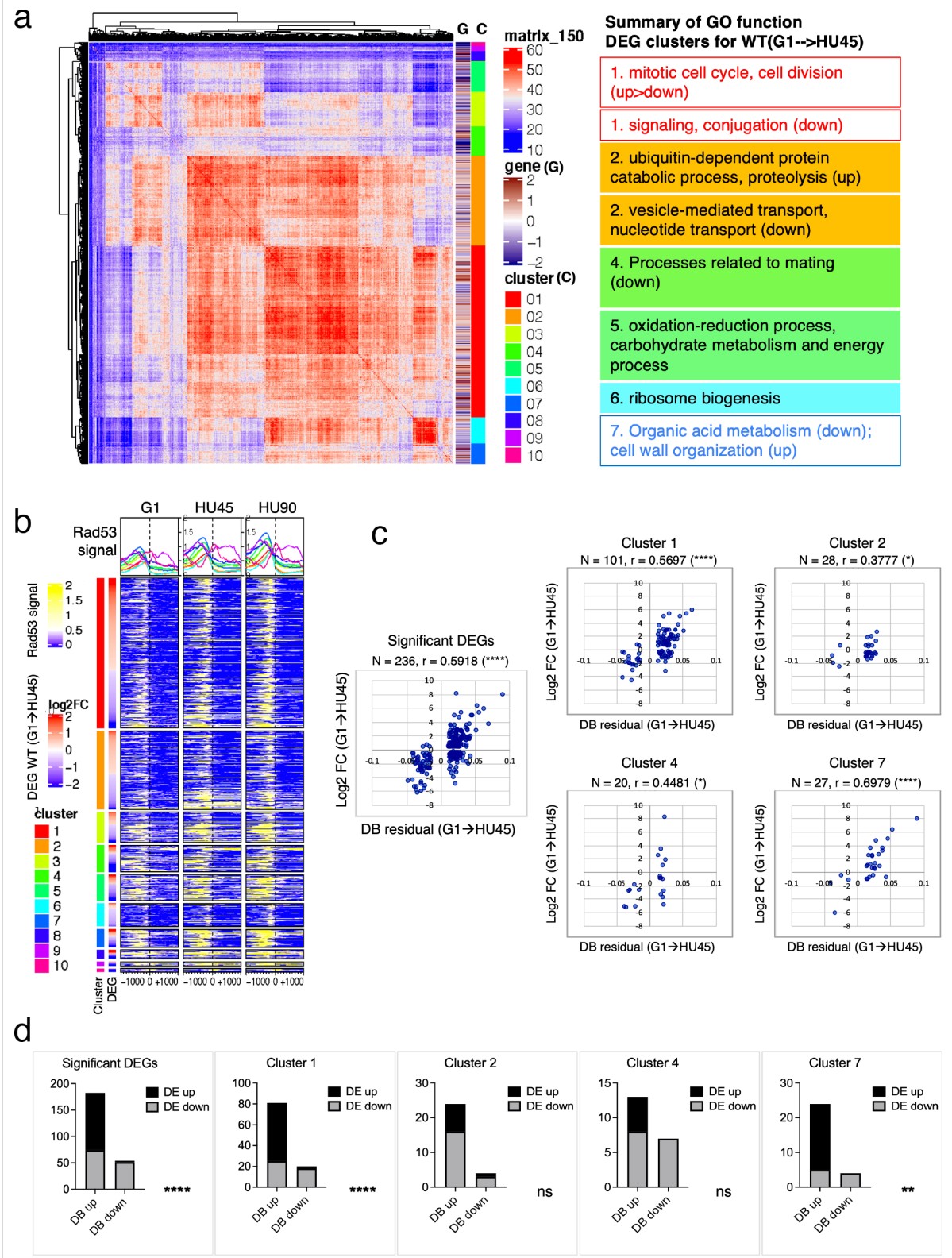

**Figure 5.** Correlation between differential binding of Rad53 at promoter and differential gene expression (DEG). (**a**) Coexpression cluster matrix for significant DEGs in *WT*(G1 → HU45). Cluster (C): color codes for DEG clusters. Gene (G): level of differential expression in log2FC. (**b**) Heatmaps of Rad53 ChIP-seq signal across 2-kb intervals centered on transcription start site (TSS) parsed by the DEG clusters in (**a**). Genes within each cluster are arranged in descending order according to the differential expression level (i.e., log2FC). (**c**) Scatter plots of Rad53-binding changes at the promoter

*Figure 5 continued on next page*

*Figure 5 continued*

against expression changes for the 236 significant DEGs in the 435 top DB overlap group (leftmost panel) and subgroups in clusters 1, 2, 4, and 7. Spearman's correlation *r* is indicated on top of each plot (****p < 0.0001; *p < 0.05). N, number of genes in the group analyzed. (**d**) Result of Fisher's exact test for association between binding changes (DB) and expression changes (DE) for groups presented in (**c**). ****p < 0.0001; **p < 0.01; ns, not significant. Examples of genes in the cluster are shown below the plot.

genes were close to late and early origins, respectively, Fisher's exact test p = 0.012). Because late origins and intermediate early origins were more active in the *mrc1Δ* mutant, it is possible that nearby gene expression was negatively affected by active DNA synthesis. In the third category, we examined the effect of relative gene–origin orientation. We found that the bias toward the downregulation was stronger when the nearby origin (<5 kb away) was in a HO orientation than in a CD orientation toward the gene (86% and 67% downregulated for HO and CD sets, respectively, Fisher's exact test p = 0.0019) (*Figure 6d*, right panels). Thus, untimely activation of DNA replication origins in the checkpoint mutants affects gene expression concomitant with Rad53 binding to gene bodies.

## Rad53-binding changes coincide with the changes in gene expression for targets of cell cycle regulators SBF and MBF

Two clusters from the gene coexpression analysis of DEGs in the *WT*(G1 → HU45) comparison showed a significant correlation between Rad53 binding and gene expression (*Figure 5c, d*, clusters 1 and 7). These two clusters contain genes that encode targets of SBF and MBF, key transcription factor complexes comprised of a shared regulatory subunit, Swi6 and the DNA-binding subunits Swi4 and Mbp1, respectively (*Breeden, 2003*). Their target genes include multiple G1- and S-phase cyclin genes, such as *PCL1*, *CLN1*, *CLN2*, *CLB5*, and *CLB6*. Evidence suggests that SBF and MBF are directly regulated by Rad53 kinase and Rad53 may regulate expression of targets of Msn4, Swi6, Swi4, and Mbp1 through Dun1-independent mechanisms (*Jaehnig et al., 2013*; *Bastos de Oliveira et al., 2012*; *Sidorova and Breeden, 2003*; *Travesa et al., 2012*). Thus, we analyzed the annotated targets of these transcription factors compiled in the *Saccharomyces* Genome Database (SGD; https://www.yeastgenome.org). Among the 81 genes that are candidate targets for both Swi4 and Swi6, 36 genes were found in the 236 DB/DEGs (*Figures 3b and 5c*) with an increase of frequency from 1.18% to 15.25% (p < 1E−15). Scatter plot comparisons of Rad53-binding and gene expression changes of these 36 genes show a clear positive correlation (*Figure 7a*, SBF top panel). Comparing with data from the checkpoint mutant samples, we found that most of these SBF target genes showed a similar profile of differential expression, from G1 to HU, in the *rad9Δ* mutant to that in *WT* (*Figure 7b*). However, in the *mrc1Δ* and *rad53^{K227A}* mutants, a subgroup of genes, for example *RNR1*, *SRL1*, and *YMR279C*, exhibited different levels of change from that in *WT* (*Figure 7b*). We also found significant enrichment for targets of MBF (targets for both Mbp1 and Swi6 in SGD annotation) and transcription factor Msn4 among the 236 DB/DEGs group, as well as positive correlations between Rad53 promoter-binding and gene expression changes in these transcription factor targets (*Figure 7a*). Noticeably, there are 19 genes in these TF target group being both targets of SBF and MBF, and 12 out of 22 Msn4 targets that are also SBF targets.

Many of the genes with decreased Rad53 binding at the promoters are mating response genes (*Figure 3b, c*, bottom panels). Therefore, the targets of Ste12, a key transcription factor activated by MAPK signaling to regulate genes involved in mating or pseudohyphal/invasive growth pathways were investigated. Of 183 potential targets of Ste12 annotated in SGD, 34 were in the 236 DB/DEGs group (*Figures 5c and 7a*). All the Ste12 targets that have decreased Rad53 binding were downregulated as cells entered S-phase. Moreover, 20 out of the 34 Ste12 targets in the top DB group showed increased Rad53 binding in HU and 11 of these 20 genes were also targets of SBF. Thus, regulation by SBF may be responsible for the correlation between increased Rad53 binding at the promoter and upregulation of these target genes.

## SBF is a key factor for recruitment of Rad53 to the promoters of its target genes under replication stress condition

To determine the contribution of various transcription regulators in recruitment of Rad53 to gene promoters, we performed Rad53 ChIP-seq experiments for *WT*, *ixr1Δ*, *swi4Δ*, and *swi6Δ* mutants (experiment TF). The *ixr1Δ* mutant was also examined because a previous investigation found that

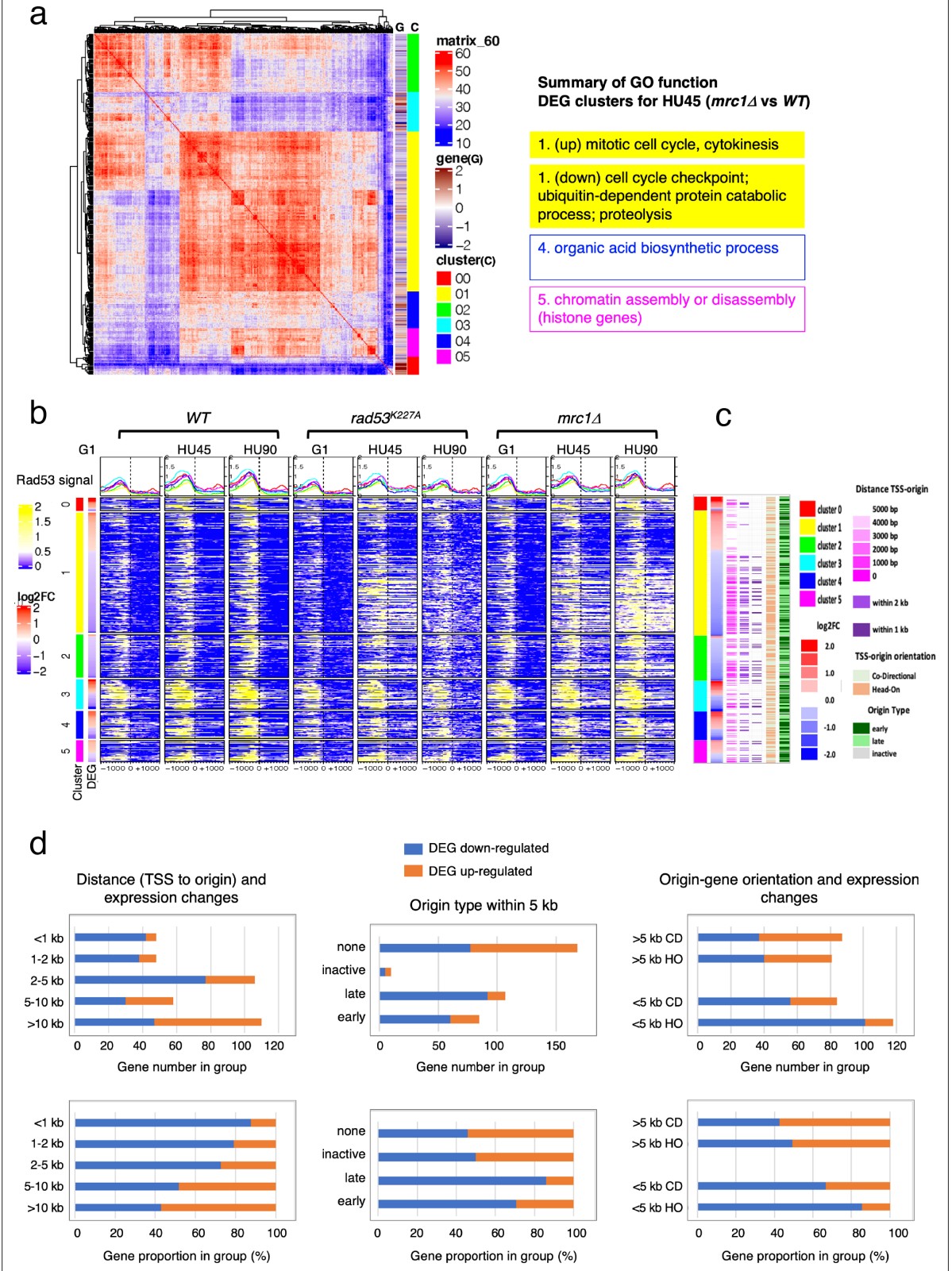

**Figure 6.** Origin-proximal differentially expressed genes (DEGs) are biased toward downregulation in the *mrc1Δ* mutant. DEGs from HU45 (*mrc1Δ* vs. *WT*) comparison were analyzed. (**a**) Coexpression cluster matrix for significant DEGs. Cluster (C): color codes for DEG clusters. Gene (G): level of differential expression in log2FC. (**b**) Heatmaps of Rad53 ChIP-seq signal across 2-kb intervals centered on transcription start site (TSS) parsed by the DEG clusters in (**a**). ChIP-seq signal in *WT*, *rad53^K227A^*, and *mrc1Δ* mutant cells at stages G1, HU45, and HU90 are shown. (**c**) Summary of gene–origin

*Figure 6 continued on next page*

*Figure 6 continued*

relation for DEGs coexpression clusters. Distance between each TSS and its nearest origin center is indicated in pink gradient as well as light purple (<2kb) and dark purple (<1kb). Relative TSS-origin orientation and origin type are indicated. (**d**) Stacked bar charts presenting number (top panels) and proportion (bottom panels) of down- and upregulated DEGs as categorized by (1) TSS to origin distance (left panels), (2) closest origin type within 5kb of TSS (middle panels; none: no origins within 5kb of TSS), and (3) origin–gene orientation (right panels; CD: co-directional; HO: head-on).

The online version of this article includes the following figure supplement(s) for figure 6:

**Figure supplement 1.** Origin-proximal differentially expressed genes (DEGs) are biased toward downregulation in the *rad53*[K227A] mutant.

Ixr1 binds to the *RNR1* promoter upon genotoxic stress and mediates Dun1-independent *RNR1* gene regulation that requires Rad53 (*Tsaponina et al., 2011*). In the scatter plot of the Rad53 signal upstream of TSSs in G1 versus HU45 from the *WT* dataset, SBF targets in the top DB (*Figure 8a*, orange/red diamonds) showed substantial deviation from the general trend (blue dots). In *swi4Δ* and *swi6Δ* mutants, the signal for all of these SBF targets collapsed toward the general trend (purple and light olive dots, *swi6Δ* and *swi4Δ*, respectively), suggesting that Rad53 signal changes at these genes depended on SBF. Analysis of *Z*-score distribution for Rad53 DB residual (G1 → HU45) also showed substantial deviation of SBF target genes from the rest of the genes in *WT* (*Figure 8b*), while in the *swi4Δ* and *swi6Δ* mutants, the deviation of the SBF targets was closer to other genes. Coverage tracks for Rad53 ChIP-seq signals showed that in the SBF mutants Rad53 binding was completely eliminated from the *TOS6* (target of SBF 6) promoter while for *PCL1* and *YOX1*, both targets of SBF, Rad53 binding did not increase in HU, in contrast to the pattern in *WT* (*Figure 8c*). Thus, SBF is important for the recruitment of Rad53 to the promoters of SBF target genes under replication stress. Interestingly, at the promoter of *RNR3*, the paralog of *RNR1*, Rad53 binding in the SBF mutants was low, even though *RNR3* is known to be target of Rfx1 and not known as target of SBF or MBF. Thus, it is possible that, in addition to Rfx1, SBF also plays a role in regulation of *RNR3* in response to replication stress.

In the *ixr1Δ* mutant, the *Z*-score distribution of Rad53 DB residual (G1 → HU45) showed a deviation similar to the level in *WT* (*Figure 8b*), and the Rad53 ChIP-seq signal for the majority of these SBF targets remained deviated from the general trend in the scatter plot. However, one clear exception was the *RNR1* gene, indicated in the close-up plots (*Figure 8a*, lower panels), whose position collapsed in all three mutants, consistent with *RNR1* being a target of Ixr1, SBF, and MBF (*de Bruin et al., 2006*; *Tsaponina et al., 2011*).

## Discussion

Previous studies have implicated Rad53 kinase in regulation of gene expression through direct interaction with various transcription regulators (*Jaehnig et al., 2013*; *Bastos de Oliveira et al., 2012*; *Sidorova and Breeden, 2003*; *Travesa et al., 2012*). Unexpectedly, we found that Rad53 bound to about 20% of the gene promoters in the yeast genome. These genes encode proteins with diverse functions, including various aspects of cell cycle, metabolism, protein modification, ion transport, cell wall organization, and cell growth. There is a general trend of increasing Rad53 level during the G1- to S-phase transition in the presence of HU (*Figures 2a and 3a*). Remarkably, Rad53 binding at promoters for genes such as *RNR1*, *RNR3*, and *TOS6* increases substantially beyond the general trend (*Figures 2c and 3c*). In contrast, Rad53 binding decreased at promoters of genes involved in response to mating pheromone as cells exited from α-factor-induced G1 arrest into the cell division cycle. The prevalent and dynamic changes in Rad53 promoter-bound levels in cells replicating in HU did not necessarily depend on checkpoint signaling activated by HU, for example, at genes like *PCL1* and *YOX1*. But in some cases, such as *RNR1*, the increase in Rad53 levels was reduced in checkpoint mutants (*Figure 2—figure supplement 4*). Furthermore, the Mrc1 checkpoint mediator, Mec1 sensor kinase, and kinase activity of Rad53 itself are not absolutely required for the recruitment of Rad53 to gene promoters (*Figure 1b* and *Figure 4—figure supplement 1*). The binding of Rad53 to numerous promoters in the yeast genome suggests a previously unappreciated level of transcription regulation that is coordinated with the many cellular processes, such as normal DNA replication, and induced cellular stresses to which Rad53 responds.

The conditions employed in this study, cell cycle entry in the presence of HU, may determine the nature of the genes that display dynamic binding of Rad53 to gene promoters. It is known that Rad53 phosphorylates transcription factors such as the SBF and MBF subunit Swi6 and the MBF corepressor

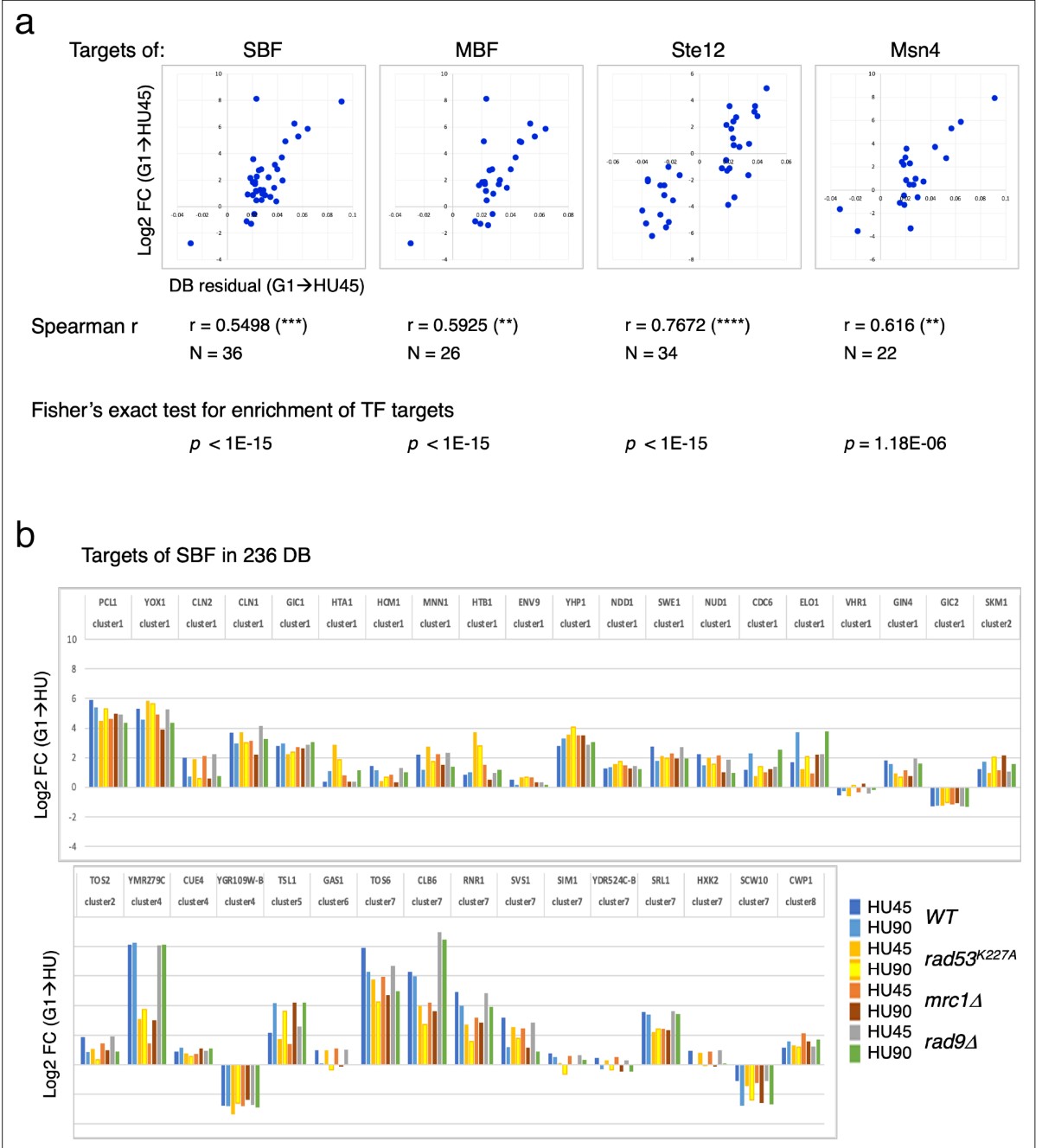

**Figure 7.** Differential binding of Rad53 at promoters and differential expression of target genes of SBF, MBF, Msn4, and Ste12. (**a**) Top panels: scatter plots of binding changes (DB residual) and expression changes (log2FC) for targets of indicated transcription regulators that are in the 236 DB/ differentially expressed genes (DEGs) group. Spearman's correlation *r* is shown under each plot (****p < 0.0001; ***p < 0.001; **p < 0.01). *N*, number of genes in the group analyzed. Result of Fisher's exact test for enrichment of each group of transcription regulator targets in the 236 DB/DEGs is also indicated below. (**b**) Profiles of differential expression in column graphs for each of the 36 SBF targets in the 236 DB/DEGs. Color-coded columns showing expression change (log2FC (G1 → HU45) and log2FC (G1 → HU90)) extracted from *WT*, *rad53^K227A^*, *mrc1Δ*, and *rad9Δ* datasets.

Nrm1 (*Bastos de Oliveira et al., 2012*; *Sidorova and Breeden, 2003*; *Travesa et al., 2012*) and that Ixr1 controls transcription of *RNR1* (*Tsaponina et al., 2011*). Removal of Swi4, Swi6 or Ixr1 reduced, and in some cases eliminated Rad53 binding to promoters of genes controlled by these transcription factors. We also found Rad53 bound to the *NRM1* promoter, which is also in the top DB group, suggesting an additional regulation of cell cycle-dependent transcription control by Rad53. Rad53 also bound to promoters of genes encoding histones and all histone genes were upregulated DEGs

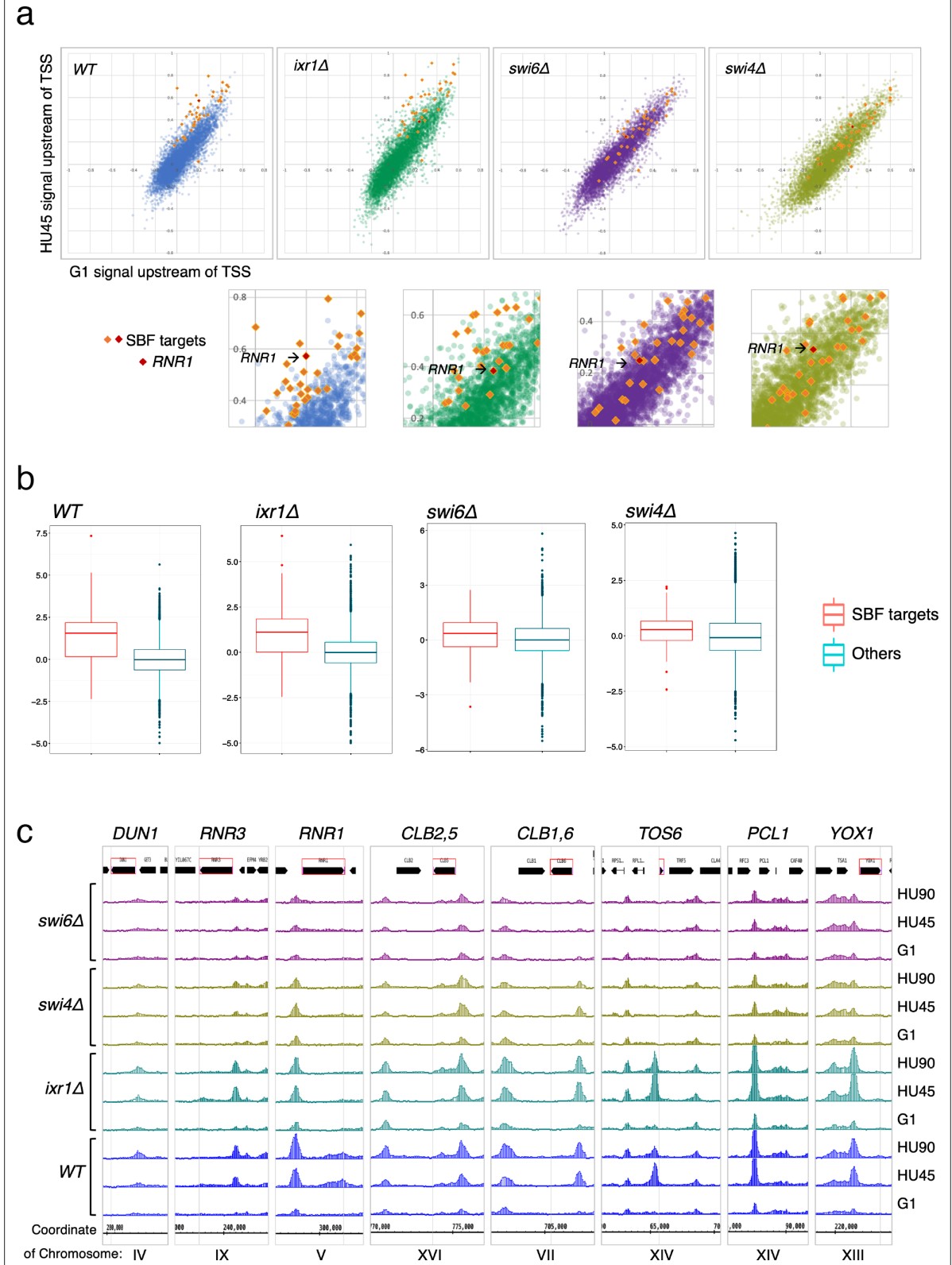

**Figure 8.** SBF plays a major role in the localization of Rad53 to the promoters of its target genes under replication stress. (**a**) Top panels: Scatter plots compare Rad53 ChIP-seq signals in G1 (*x*-axis) and HU45 (*y*-axis) at 500-bp intervals upstream of transcription start sites (TSSs) for all genes in *WT*, *ixr1Δ*, *swi4Δ*, and *swi6Δ* mutants. Signals for SBF targets found in the 435 top DB overlap are shown as orange or red diamond (*RNR1* in red diamond). Signals for the rest of genes are shown as dots in color blue (*WT*), green (*ixr1Δ*), purple (*swi6Δ*), and light olive (*swi4Δ*). Bottom panels: Close-up for specific area

*Figure 8 continued on next page*

*Figure 8 continued*

from above panels. (**b**) Box plots showing *Z*-score distribution for DB residuals (G1 → HU45) for genes that are annotated as targets for both Swi4 and Swi6 (SBF targets) and genes that are otherwise (Others) in *WT*, *ixr1Δ*, *swi4Δ*, and *swi6Δ* mutants (p value by Wilcoxon rank sum test: *WT*, 2.2e-16; *ixr1Δ*, 3.174e−12; *swi4Δ*, 0.001608; *swi6Δ*, 0.002857). (**c**) Coverage tracks showing distribution of Rad53 ChIP-seq signal near selected top DB genes in *WT*, *ixr1Δ*, *swi4Δ*, and *swi6Δ* mutants at stages of G1, HU45, and HU90.

in the HU45 (*rad53*$^{K227A}$ vs. *WT*) comparison (***Figure 6—figure supplement 1***), suggesting that, in addition to its known role in histone degradation (***Gunjan and Verreault, 2003***), Rad53 may control histone level through gene expression. This is consistent with previous findings that Rad53 targets Yta7 (***Smolka et al., 2006***), which interacts with FACT to regulate histone gene expression and inhibits Spt21$^{NPAT}$-regulated histone genes expression (***Bruhn et al., 2020***; ***Gradolatto et al., 2008***). In the absence of Rad53 protein, histone levels become elevated, causing global effects on gene expression (***Bruhn et al., 2020***; ***Tsaponina et al., 2011***).

Our data are consistent with the possibility that the Rad53 protein contributes to the transcriptional regulation as a structural component by binding directly to promoters or transcriptions factors bound to these promoters, as previously suggested for several MAP kinases (***Alepuz et al., 2001***; ***Kim et al., 2008***; ***Sanz et al., 2018***). Like the stress-induced kinase Hog1, Rad53 binding to promoters may be dynamic in other stress conditions, which is under investigation. A major unanswered question is how Rad53 binds to so many diverse promoter sites. We suggest that it may bind DNA directly or to a common factor that is present at the promoters of Rad53-bound genes. Alternatively, certain chromatin features common to these promoters may facilitate Rad53 recruitment. It remains possible that, in addition to transcriptional regulation, Rad53 may be recruited to promoter region preemptively to ensure integrity of vulnerable chromatin associated with active transcription. Pre-targeting repair complexes to open chromatin where they are poised for lesion recognition and repair has been reported in human cells (***Bacolla et al., 2021***).

In cells, the same DNA template is used for both replication and transcription, potentially creating conflicts between replication and transcription that can lead to detrimental effects on genome stability and cell viability (***Hamperl and Cimprich, 2016***). Thus, it is important that cells employ mechanisms to avoid, tolerate and resolve such conflicts. It is known that late-replicating genes are tethered to the nuclear pore complexes in the nuclear periphery and checkpoint signaling, including Rad53 kinase, is required for preventing topological impediments for replication fork progression (***Bermejo et al., 2011***; ***Hamperl and Cimprich, 2016***). Moreover, during normal replication, Mec1 may locally activate Rad53 to deal with difficult to replicate loci or regions of replication–transcription conflict without triggering full blown checkpoint activation (***Bastos de Oliveira et al., 2015***).

Eukaryotic cells initiate DNA synthesis in a temporally controlled manner from multiple replication origins to ensure efficient duplication of the genome (***Bell and Labib, 2016***; ***Renard-Guillet et al., 2014***). One of the main consequences of checkpoint activation is inhibition of late origin firing (***Tercero et al., 2003***). However, in the checkpoint mutants, these late origins become active and we found that Rad53 was recruited to the body of origin-proximal genes. Concomitantly, gene expression of these genes was reduced, perhaps mediated by recruitment of Rad53. While checkpoint signaling pathways are known to regulate gene expression, it is possible that untimely activation of the late origins results in a replication–transcription conflict and contribute to downregulation of genes in these checkpoint mutants. We suggest that the normal temporal order of replication of genes in the genome throughout S-phase, and possibly the order of transcription of genes, have evolved to prevent conflicts between replication and transcription, which is particularly important in a gene dense genome such as *S. cerevisiae*.

# Materials and methods

**Key resources table**

| Reagent type (species) or resource | Designation | Source or reference | Identifiers | Additional information |
|---|---|---|---|---|
| Strain, strain background (*Saccharomyces cerevisiae*) | YS2571 | doi:10.1073/ pnas.1404063111 | | *MATa bar1Δ::TRP1 URA3::BrdU-Inc ade2-1 can1-100 his3-11,–15 leu2-3,112 trp1-1 ura3-1* |

*Continued on next page*

| Reagent type (species) or resource | Designation | Source or reference | Identifiers | Additional information |
|---|---|---|---|---|
| Strain, strain background (*Saccharomyces cerevisiae*) | YS3110 | This paper | | *MATa rad53^K227A::KanMX4 bar1Δ::TRP1 URA3::BrdU-Inc ade2-1 can1-100 his3-11,–15 leu2-3,112 trp1-1 ura3-1* |
| Strain, strain background (*Saccharomyces cerevisiae*) | YS3285 | This paper | | *MATa mrc1Δ::KanMX4 bar1Δ::TRP1 URA3::BrdU-Inc ade2-1 can1-100 his3-11,–15 leu2-3,112 trp1-1 ura3-1* |
| Strain, strain background (*Saccharomyces cerevisiae*) | YS3382 | This paper | | *MATa rad9Δ::HIS3 bar1Δ::TRP1 URA3::BrdU-Inc ade2-1 can1-100 his3-11,–15 leu2-3,112 trp1-1 ura3-1* |
| Strain, strain background (*Saccharomyces cerevisiae*) | YS3388 | This paper | | *MATa ixr1Δ::HIS3 bar1Δ::TRP1 URA3::BrdU-Inc ade2-1 can1-100 his3-11,–15 leu2-3,112 trp1-1 ura3-1* |
| Strain, strain background (*Saccharomyces cerevisiae*) | YS3401 | This paper | | *MATa swi4Δ::HIS3 bar1Δ::TRP1 URA3::BrdU-Inc ade2-1 can1-100 his3-11,–15 leu2-3,112 trp1-1 ura3-1* |
| Strain, strain background (*Saccharomyces cerevisiae*) | YS3406 | This paper | | *MATa swi6Δ::HIS3 bar1Δ::TRP1 URA3::BrdU-Inc ade2-1 can1-100 his3-11,–15 leu2-3,112 trp1-1 ura3-1* |
| Strain, strain background (*Saccharomyces cerevisiae*) | YS2828 | doi:10.1101/gr.195248.115 | | *MATa URA3::BrdU-Inc ade2-1 can1-100 his3-11,–15 leu2-3,112 trp1-1 ura3-1* |
| Strain, strain background (*Saccharomyces cerevisiae*) | YS3066 | doi:10.1101/gr.195248.116 | | *MATa sml1Δ::HIS3 URA3::BrdU-Inc ade2-1 can1-100 his3-11,–15 leu2-3,112 trp1-1 ura3-1* |
| Strain, strain background (*Saccharomyces cerevisiae*) | YS3075 | doi:10.1101/gr.195248.117 | | *MATa mec1Δ::TRP1 sml1Δ::HIS3 URA3::BrdU-Inc ade2-1 can1-100 his3-11,–15 leu2-3,112 trp1-1 ura3-1* |
| Strain, strain background (*Saccharomyces cerevisiae*) | YS3077 | doi:10.1101/gr.195248.118 | | *MATa rad53Δ::KanMX sml1Δ::HIS3 URA3::BrdU-Inc ade2-1 can1-100 his3-11,–15 leu2-3,112 trp1-1 ura3-1* |
| Antibody | Anti-Rad53 antibody | Abcam | Cat# ab104232, RRID:AB_2687603 | |
| Antibody | Anti-Histone H2A (phospho S129) antibody | Abcam | Cat# ab15083, RRID:AB_301630 | |
| Antibody | Anti-Cdc45 antibody (polyclonal CS1485) | doi:10.1016/j.molcel.2006.07.033 | | |
| Antibody | Anti-Orc6 antibody (monoclonal SB49) | Other | | Stillman lab |
| Antibody | Anti Sml1 \| Suppressor of Mec1 lethality antibody | Agrisera | Cat# AS10 847 | |
| Peptide, recombinant protein | α-Factor | Other | | WHWLQLKPGQPMY |
| Commercial assay or kit | TruSeq ChIP Library Preparation Kit | Illumina | Cat# IP-202-1012, IP-202-1024 | |
| Commercial assay or kit | TruSeq stranded mRNA library preparation kit | Illumina | Cat# 20020594 | |
| Commercial assay or kit | MinElute PCR purification kit | Qiagen | Cat# 28004 | |
| Chemical compound, drug | Hydroxyurea | Sigma | H8627-25G | |
| Software, algorithm | GraphPad Prism 9 | GraphPad | RRID:SCR_002798 | |
| Software, algorithm | Bowtie | doi:10.1002/0471250953.bi1107s32 | RRID:SCR_005476 | |
| Software, algorithm | bamCoverage | doi:10.1093/nar/gku365 | | |

| Reagent type (species) or resource | Designation | Source or reference | Identifiers | Additional information |
|---|---|---|---|---|
| Software, algorithm | STAR | doi:10.1093/bioinformatics/bts635 | | |
| Software, algorithm | MACS2 | | RRID:SCR_013291 | |
| Software, algorithm | ChIPpeakAnno | | RRID:SCR_012828 | |
| Software, algorithm | DescTools | Andri Signorell et mult. al. (2021) | | DescTools: Tools for descriptive statistics. R package version 0.99.41, 598 https://cran.r-project.org/package=DescTools |

## Yeast strains and methods

Yeast strains generated in this study were derived from W303-1a (*MATa ade2-1 can1-100 his3-11,15 leu2-3,112 trp1-1 ura3-1*) and are described in *Table 1*. All the yeast strains used for the whole-genome DNA replication profile analyses have a copy of the BrdU-Inc cassette inserted into the URA3 locus (*Viggiani and Aparicio, 2006*). For G1 arrest of *bar1Δ* strains, exponentially growing yeast cells (~$10^7$ cell/ml) in YPD were synchronized in G1 with 25 ng/ml of α-factor for 150 min at 30°C. For G1 arrest of *BAR1* strains, exponentially growing cells were grown in normal YPD, then transferred into YPD (pH 3.9), grown to ~$10^7$ cell/ml, and then synchronized in G1 with three doses of α-factor at 2 μg/ml at 0-, 50-, and 100 min time point at 30°C. Cells were collected at 150 min for release. To release from G1 arrest, cells were collected by filtration and promptly washed twice on the filter using one culture volume of $H_2O$ and then resuspended into YPD medium containing 0.2 mg/ml pronase E (Sigma).

## Protein sample preparation and immunoblot analysis

TCA extraction of yeast proteins was as described previously (*Sheu et al., 2014*). For immunoblot analysis, protein samples were fractionated by sodium dodecyl sulfate–polyacrylamide gel electrophoresis (SDS–PAGE) and transferred to a nitrocellulose membrane. Immunoblot analyses for Orc6 (SB49), Rad53 (ab104232, Abcam), γ-H2A (ab15083, Abcam), and Sml1 (AS10 847, Agrisera) were performed as described (*Sheu et al., 2016*; *Sheu et al., 2014*).

## Isolation and preparation of DNA for whole-genome replication profile analysis

Modified protocol based on previously described (*Sheu et al., 2016*; *Sheu et al., 2014*). Briefly, yeast cells were synchronized in G1 with α-factor and released into medium containing 0.2 mg/ml pronase E, 0.5 mM 5-ethynyl-2'-deoxyuridine (EdU) with or without addition of 200 mM HU as indicated in the main text. At the indicated time point, cells were collected for preparation of genomic DNA. The

**Table 1.** Yeast strains used in this study.

| Strain | Genotype | Source |
|---|---|---|
| YS2571 | *MATa bar1Δ::TRP1 URA3::BrdU-Inc ade2-1 can1-100 his3-11,–15 leu2-3,112 trp1-1 ura3-1* | *Sheu et al., 2014* |
| YS3110 | *MATa rad53^{K227A}::KanMX4 bar1Δ::TRP1 URA3::BrdU-Inc ade2-1 can1-100 his3-11,–15 leu2-3,112 trp1-1 ura3-1* | This study |
| YS3285 | *MATa mrc1Δ::KanMX4 bar1Δ::TRP1 URA3::BrdU-Inc ade2-1 can1-100 his3-11,–15 leu2-3,112 trp1-1 ura3-1* | This study |
| YS3382 | *MATa rad9Δ::HIS3 bar1Δ::TRP1 URA3::BrdU-Inc ade2-1 can1-100 his3-11,–15 leu2-3,112 trp1-1 ura3-1* | This study |
| YS3388 | *MATa ixr1Δ::HIS3 bar1Δ::TRP1 URA3::BrdU-Inc ade2-1 can1-100 his3-11,–15 leu2-3,112 trp1-1 ura3-1* | This study |
| YS3401 | *MATa swi4Δ::HIS3 bar1Δ::TRP1 URA3::BrdU-Inc ade2-1 can1-100 his3-11,–15 leu2-3,112 trp1-1 ura3-1* | This study |
| YS3406 | *MATa swi6Δ::HIS3 bar1Δ::TRP1 URA3::BrdU-Inc ade2-1 can1-100 his3-11,–15 leu2-3,112 trp1-1 ura3-1* | This study |
| YS2828 | *MATa URA3::BrdU-Inc ade2-1 can1-100 his3-11,–15 leu2-3,112 trp1-1 ura3-1* | *Sheu et al., 2016* |
| YS3066 | *MATa sml1Δ::HIS3 URA3::BrdU-Inc ade2-1 can1-100 his3-11,–15 leu2-3,112 trp1-1 ura3-1* | *Sheu et al., 2016* |
| YS3075 | *MATa mec1Δ::TRP1 sml1Δ::HIS3 URA3::BrdU-Inc ade2-1 can1-100 his3-11,–15 leu2-3,112 trp1-1 ura3-1* | *Sheu et al., 2016* |
| YS3077 | *MATa rad53Δ::KanMX sml1Δ::HIS3 URA3::BrdU-Inc ade2-1 can1-100 his3-11,–15 leu2-3,112 trp1-1 ura3-1* | *Sheu et al., 2016* |

genomic DNA were fragmented, biotinylated, and then purified. Libraries for Illumina sequencing were constructed using TruSeq ChIP Library Preparation Kit (Illumina). Libraries were pooled and submitted for 50-bp paired-end sequencing.

## Sample preparation for ChIP-seq

Chromatin immunoprecipitation (ChIP) was performed as described (*Behrouzi et al., 2016*) with modification. About $10^9$ synchronized yeast cells were fixed with 1% formaldehyde for 15min at room temperature (RT), then quenched with 130mM glycine for 5min at RT, harvested by centrifugation, washed twice with tris-buffered saline (50mM Tris–HCl pH 7.6, 150mM NaCl), and flash frozen. Cell pellets were resuspended in 600µl lysis buffer (50mM HEPES–KOH pH 7.5, 150mM NaCl, 1mM ethylenediaminetetraacetic acid, 1% Triton X-100, 0.1% Na-deoxycholate, 0.1% SDS, 1mM phenyl-methylsulfonyl fluoride, protease inhibitor tablet (Roche)), and disrupted by bead beating using multi-tube vortex (Multi-Tube Vortexer, Baxter Scientific Products) for 12–15 cycles of 30s vortex at maximum intensity. Cell extracts were collected and sonicated using Bioruptor (UCD-200, Diagenode) for 38 cycles of pulse for 30s 'ON', 30s 'OFF' at amplitude setting High (H). The extract was centrifuged for 5min at 14,000rpm. The soluble chromatin was used for IP.

Antibodies against Cdc45 (CS1485, this lab *Sheu and Stillman, 2006*), Rad53 (ab104232, Abcam), γ-H2A (ab15083, Abcam) was preincubated with 1:1 mixture of washed Dynabeads Protein A and G (1002D and 1004D, Invitrogen) for more than 30min at RT and washed twice with lysis buffer to remove unbound antibodies. For each immunoprecipitation, 80µl antibody-coupled Dynabeads was added to soluble chromatin. Samples were incubated overnight at 4°C with rotation, after which the beads were collected on magnetic stands, and washed three times with 1ml lysis buffer and once with 1ml Tris-EDTA (50 mM Tris.–HCl pH 8.0, 10 mM EDTA), and eluted with 250µl preheated buffer (50mM Tris–HCl pH 8.0, 10mM EDTA, 1% SDS) at 65°C for 15min. Immunoprecipitated samples were incubated overnight at 65°C to reverse crosslink, and treated with 50µg RNase A at 37°C for 1hr. Then 5µl proteinase K (Roche) in 20mg/ml stock was added and incubation was continued at 55°C for 1hr. Samples were purified using MinElute PCR purification kit (Qiagen). Libraries for Illumina sequencing were constructed using TruSeq ChIP Library Preparation Kit (IP-202-1012 and IP-202-1024, Illumina).

## Sample preparation for RNA-seq

About $2–3 \times 10^8$ flash-frozen yeast cells were resuspended in Trizol (cell pellet:Trizol = 1:10) and vortexed for 15 s and incubated at 25°C for 5 min. Subsequently, 200 µl chloroform was added per 1 ml of Trizol–cell suspension and samples were vortexed for 15 s, incubated at RT for 5 min and centrifuged to recover the aqueous layer. The RNA in the aqueous layer were further purified and concentrated using PureLink Column (12183018A, Invitrogen). The RNA was eluted in 50 µl and store at −20°C if not used immediately. Store at −80°C for long term. Paired-end RNA-seq libraries were prepared using TruSeq stranded mRNA library preparation kit (20020594, Illumina).

## Generation of coverage tracks using the Galaxy platform

For visualization of read coverage in the Integrated Genome Browser (*Freese et al., 2016*), the coverage tracks were generated using the Galaxy platform maintained by the Bioinformatics Shared Resource (BSR) of Cold Spring Harbor Lab. The paired-end reads from each library were trimmed to 31 bases and mapped to sacCer3 genome using Bowtie (*Langmead, 2010*). The coverage track of mapped reads was then generated using bamCoverage (*Ramírez et al., 2014*) with normalization to 1× genome.

## Definition of the origin types

Based on the BamCoverage output for EdU signal in *WT*, *rad53^K227A^*, and *mrc1Δ*, we categorized 829 origins listed in the oriDB database (*Siow et al., 2012*). We define the early origins as the one whose signal at the first time point is larger than 2. The late origins are extracted from the rest of the origins if the average signal value at the later time point is larger than 2 in *rad53^K227A^* and *mrc1Δ* mutants. Among the 829 entries in oriDB, we defined 521 as active origins (with EdU signal in *WT* or checkpoint mutants *rad53^K227A^* and *mrc1Δ*), in which 256 was categorized as early origins (with EdU signal in *WT*) and 265 as late origins (with signal in checkpoint mutants but not in *WT*). The remaining 308 entries do not have significant signal under our condition and were deemed inactive origins.

## Computational analysis of sequence data

The sequenced reads were trimmed by cutadapt with an option of 'nextseq-trim', then aligned by STAR (*Dobin et al., 2013*) in a paired-end mode to the sacCer3 genome masked at repetitive regions. The gene structure is referred from SGD reference genome annotation R64.1.1 as of October 2018. For RNA-seq quantification analysis, the total counts of aligned reads were computed for each gene by applying 'GeneCounts' mode. For ChIP-seq quantification analysis, the reads were mapped using the same pipeline. We also confirmed the mapped reads found in ChIP-seq data do not span a long range (the median span is from 169.5 to 329, 90% paired-end reads are aligned within less than 6000-bp window), suggesting that STAR spliced alignment do not affect the alignment results. Additionally, peak calling was done by MACS2 in a narrow peak mode. Distribution of Rad53 ChIP-seq peaks was computed using ChIPpeakAnno.

The Gini indexes were calculated from Lorenz curves using Rad53 ChIP-seq datasets and published ChIP-Seq data for Swi6 (SRX360900 <https://www.ncbi.nlm.nih.gov/sra/SRX360900%5baccn%5d>: GSM1241092: swi6 DMSO illumina; *S. cerevisiae*; ChIP-seq), using DescTools (Andri Signorell et mult. al. (2021). DescTools: Tools for descriptive statistics. R package version 0.99.41, https://cran.r-project.org/package=DescTools).

## Gene expression analysis

DEGs and their p values were computed for each pair of the cases by nbinomWaldTest after size factor normalization using DESeq2 (*Love et al., 2014*). Using the list of DEGs, GO and KEGG enrichment analyses were performed via Pathview library. ClusterProfiler was applied to visualize fold changes of DEGs in each KEGG pathway. Coexpression analysis of significant DEGs was further performed based on coexpression network constructed in CoCoCoNet (*Lee et al., 2020*). CoCoCoNet has established the coexpression matrix of Spearman's correlation ranking based on 2690 samples downloaded from SRA database. We carried out clustering for the correlation matrix downloaded from CoCoCoNet (yeast_metaAggnet) by dynamicTreeCut in R (or hierarchical clustering) to obtain at most 10 clusters. The enrichment analysis for the gene set of each cluster was performed in the same way with RNA-seq analysis.

## ChIP-seq signal normalization

For ChIP-seq signal normalization, two different methods were applied to different types of analysis. For ChIP-seq residual analysis, we used simple normalization. In this process, each case sample was compared with the corresponding control sample of DNA input to compute log2 fold changes within each 25-bp window reciprocally scaled by multiplying the total read counts of another sample. Then, the average of fold changes was computed for each duplicate. For ChIP-seq heatmap analysis, we employed the origin-aware normalization to account for the higher background around origin region as a result of DNA replication. In the origin-aware normalization, the same computation used in simple normalization, or log2 fold change with scaling by the total read count, is independently applied for the region proximal to the origins and others. For the heatmap presented in this paper, the origin-proximal region is defined as the region within 5000-bp upstream and downstream.

## Heatmap analyses at origins and TSS

After the average fold change computation and normalization from ChIP-seq signals, the signal strength is visualized around the target regions such as TSSs and replication origins are extracted using normalizeToMatrix function in EnrichedHeatmap (window size is 25 bp and average mode is w0). We ordered heatmaps to examine a different signal enrichment pattern for the characteristics of each origin or gene. For the heatmap row of each origin is ordered by the assigned replication timing for ChIP-seq signals around replication origins. The replication time for the origins are annotated with the replication timing data published previously (*Yabuki et al., 2002*). From the estimated replication time for each 1000-bp window, we extracted the closest window from the center of each replication origin and assigned it as the representative replication timing if their distance is no more than 5000 bp. Early and late origins groups are categorized according to the definition of the origin types using the replication profile data from this study. The final set of the replication origins used in the heatmap analysis are obtained after filtering out the replication origins overlapped with any of

238 hyper-ChIPable regions defined in the previous study (*Teytelman et al., 2013*). In total, 167 early and 231 late origins pass this filter and are used in the heatmaps analysis in this study. For heatmaps of the ChIP-seq signals around TSS, we ordered genes based on RNA-seq fold changes for all DEGs or per coexpression cluster of DEGs based on gene coexpression network constructed in CoCoCoNet (*Lee et al., 2020*).

## ChIP-seq residual analysis

To detect the time-dependent increase or decrease of Rad53-binding signals, we first focused on the 500-bp window upstream from each TSS (defined as promoter region) and computed the sum of the fold change signals estimated for each 25-bp window scaled by the window size as an activity of Rad53 binding for each gene. The overall activity scores are varied for each time point probably because of the different Rad53 protein level or other batch-specific reasons. To adjust such sample-specific differences for a fair comparison, a linear regression is applied for the activity scores of all genes between G1 and other time points HU45 and HU90 using lm function in R. Then we selected top genes showing the deviated signals from the overall tendency according to the absolute residual values between the actual and predicted values, excluding the genes with signal value lower than −0.075 after scaling the maximal signal to 1. Top 1000 genes with the highest absolute residual values were selected from two independent experiments. The 435 DB genes identified in both experiments (435 top DB overlap) were selected for further analysis.

## Acknowledgements

This research was supported by NIH grants R01GM45436 and R01LM012736 and a gift from the Goldring Family Foundation. The Cold Spring Harbor Laboratory Cancer Center supported core research resources (P30-CA045508). RKK was supported by Uehara Memorial Foundation Postdoctoral Fellowship.

## Additional information

### Competing interests

Bruce Stillman: Reviewing editor, *eLife*. The other authors declare that no competing interests exist.

### Funding

| Funder | Grant reference number | Author |
| --- | --- | --- |
| Uehara Memorial Foundation | Postdoctoral Fellowship | Risa Karakida Kawaguchi |
| National Institute of General Medical Sciences | GM45436 | Bruce Stillman |
| National Cancer Institute | P30-CA045508 | Bruce Stillman |
| National Library of Medicine | R01LM012736 | Jesse Gillis |
| Goldring Family Foundation | | Bruce Stillman |

The funders had no role in study design, data collection, and interpretation, or the decision to submit the work for publication.

### Author contributions

Yi-Jun Sheu, Conceptualization, Resources, Data curation, Formal analysis, Validation, Investigation, Visualization, Methodology, Writing - original draft, Writing – review and editing; Risa Karakida Kawaguchi, Data curation, Software, Formal analysis, Investigation, Writing – review and editing; Jesse Gillis, Conceptualization, Data curation, Software, Formal analysis, Supervision, Funding acquisition,

Investigation, Methodology, Writing – review and editing; Bruce Stillman, Conceptualization, Data curation, Formal analysis, Supervision, Funding acquisition, Investigation, Methodology, Writing - original draft, Project administration, Writing – review and editing

### Author ORCIDs
Yi-Jun Sheu ⓘ http://orcid.org/0000-0003-1612-5708
Jesse Gillis ⓘ http://orcid.org/0000-0002-0936-9774
Bruce Stillman ⓘ http://orcid.org/0000-0002-9453-4091

### Ethics
This study used animals for antibody production and all studies were performed in strict accordance with the recommendations in the Guide for the Care and Use of Laboratory Animals of the National Institutes of Health. All of the animals were handled according to approved institutional animal care and use committee (IACUC) protocols (#20-17-14-11-8-05-02-18) of Cold Spring Harbor Laboratory.

### Decision letter and Author response
Decision letter https://doi.org/10.7554/eLife.84320.sa1
Author response https://doi.org/10.7554/eLife.84320.sa2

---

## Additional files

### Supplementary files
• MDAR checklist

### Data availability
Sequencing data have been deposited in Dryad (https://doi.org/10.5061/dryad.tx95x69xv). The scripts used in this study including RNA-seq, ChIP-seq, and co-expression analysis are available at https://github.com/carushi/yeast_coexp_analysis (copy archived at swh:1:rev:3c890db7960a7665c34490593196732d5c3ad789).

The following dataset was generated:

| Author(s) | Year | Dataset title | Dataset URL | Database and Identifier |
|---|---|---|---|---|
| Stillman B, Gillis J, Kawaguchi R, Sheu Y | 2022 | Prevalent and dynamic binding of the cell cycle checkpoint kinase Rad53 to gene promoters | https://doi.org/10.5061/dryad.tx95x69xv | Dryad Digital Repository, 10.5061/dryad.tx95x69xv |

The following previously published datasets were used:

| Author(s) | Year | Dataset title | Dataset URL | Database and Identifier |
|---|---|---|---|---|
| Teytelman L, Thurtle DM, Rine J, van Oudenaarden A | 2013 | Sir2 input for Highly Expressed are Vulnerable to Misleading ChIP Localization of Multiple Unrelated Proteins | https://www.ncbi.nlm.nih.gov/sra/SRX362242 | NCBI Sequence Read Archive, SRX362242 |
| Park D, Lee Y, Bhupindersingh G, Iyer VR | 2013 | Widespread Misinterpretable ChIP-seq Bias in Yeast | https://www.ncbi.nlm.nih.gov/geo/query/acc.cgi?acc=GSE51251 | NCBI Gene Expression Omnibus, GSE51251 |

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
