## [Editor Report]

The unexpected localization of a cell cycle checkpoint kinase, Rad53, to promoters in response to replication stress suggests that Rad53 may help coordinate transcription in response to disrupted replication. This work will be of interest to those interested in the interplay between genome stability and gene expression.

---

## [Decision Letter]

**Decision letter after peer review:**

[Editors’ note: the authors submitted for reconsideration following the decision after peer review. What follows is the decision letter after the first round of review.]

Thank you for submitting the paper "Prevalent and Dynamic Binding of the Cell Cycle Checkpoint Kinase Rad53 to Gene Promoters" for consideration by *eLife*. Your article has been reviewed by 3 peer reviewers, and the evaluation has been overseen by a Reviewing Editor and a Senior Editor. The following individual involved in review of your submission has agreed to reveal their identity: Miles A Pufall (Reviewer #2).

Comments to the Authors:

We are sorry to say that, after consultation with the reviewers, we have decided that this work will not be considered further for publication by *eLife*.

After discussing the manuscript, the reviewers and the reviewing editor concluded that while the paper reports a series of interesting observations related to Rad53 binding to gene promoters independent of checkpoint signaling, the reviewers found the main message of the paper to be unclear. The paper starts with an attempt to look at RAD53 binding to replicons, but finds binding of RAD53 to promoter sites in G1 and in the absence of replication stress. Despite this intriguing observation, the results are analyzed with respect to stressed replication. As a result, the paper does not address reasons or consequences of the observed phenotype other than transcription/replication collisions.

From a technical point of view, all reviewers were concerned with the lack of statistics to draw inferences regarding the central question: Does Rad53 coordinate both replication and transcription in response to stress, and do the mutants disrupt this coordination. While it is often useful, to note a trend that is worth exploring, where numbers are available the validity of these inferences must be drawn to strengthen the conclusions.

*Reviewer #1:*

Strength: Upon first reading of the abstract, there was some concern that the observed Rad53 ChIP-seq at promoters could be related to the "expression bias" previously reported in ChIP-seq data sets, where false peaks were observed at some highly-expressed genes, even for exogeneous non-DNA binding proteins. The authors' analyses, especially lines 229-246, alleviate this concern.

Oftentimes throughout the manuscript the authors mention trends in the data, but without providing actual numbers. E.g. on line 180: "we noticed many Rad53 peaks […] and many of these localized […]"; providing numbers would be helpful for readers to understand the scale of the observed trends. The authors refer to specific figures, but those figures do not contain any numbers either.

As another example, at line 198, the authors say "Additional genes show increased Rad53 binding […], but at other promoters Rad53 binding decreases during the same time course […]. However, at most genes Rad53 remains constant." Numbers would be helpful here. Also, how were the increases and decreases called? What statistical tests and cutoffs were used to make these calls?

Same questions for page 6, lines 216-217. And also line 221, "numerous gene promoters" – could this be expressed as a fraction of all gene promoters in yeast?

The authors used "residual analysis in WT" to identify "the top differentially binding (DB) genes" in each set. Why residual analysis and not an established tool for differential binding, such as DEseq? And how were the different experiments in each set combined? For example, the TP set has 3 transcription factor null mutants and one WT condition. Comparing each mutant to the WT will gives three sets of differentially bound gene promoters; how are these combined? Is the direction of the change always the same across the 3 mutants? If not, what is reported in Figure 4b? What exactly is the y-axis in this figure panel?

The authors continue to say "Many of these genes encode proteins involved in cell cycle progression and cell growth". How was this determined? Was a statistical analysis (like GSEA) performed?

The relevance of the comparison between Rad53 ChIP-seq and Swi6 ChIP-seq is not entire clear. Did the authors mean to illustrate that although Rad53 binds to promoters, its pattern of binding is different from that of transcription factors? I.e. Rad53 binds a large set of promoters, but the range of binding signals is generally narrow, while transcription factors tend to bind strongly but to fewer promoters? If this was the intended purpose of the analysis (presented at lines 221-227), why was Swi6 selected? Would other cell-cycle transcription factors also be relevant for this type of analysis?

The comparisons between Rad53 binding and gene expression (line 248-284) show a clear positive correlation, especially in certain gene subsets/clusters. Further analyses, focused on targets of SBF, MBF, Msn4 and Ste12, confirm these correlations on subsets of genes targeted by these transcription factors. But it is unclear, to this reviewer, whether the authors conclusion that "regulation by SBF appears to be responsible for the correlation between increased Rad53 binding at the promoter and up-regulation of these target genes" is truly supported by the data. The correlations and overlaps are compelling, this is true, but are these significant and do they point to causality?

In Figure 8b, the authors highlight a few SBF targets that show "significant deviation" from the global trend. (How what the significance assessed?) But these are just the SBF targets in the "Top DB" set, i.e. in differentially bound gene promoters. Since this is how they were selected, shouldn't we expect them to deviate from the global trend? It is not clear how this finding shows that "Rad53 signal changes at these genes depends on SBF". This section (lines 364-283) continues with some examples of specific genes and their Rad53 promoter binding chances. But it is not clear what the general conclusion is according to these data.

One aspect that makes the manuscript difficult to read is that there are many observations reported throughout the paper, but it is not always clear how they relate to the main message of the paper. E.g. line 132: "the replication fork collapse was more severe in the absence of Rad53 kinase compared to the absence of checkpoint signaling in the mrcD mutant". This is an interesting observation, but what is the relevance for the main message of the paper?

As another example, in the paragraph at line 60 (Introduction), the authors talk about Rfx1 and Ixr1 transcription factors, and the RNR genes. But the relevance of this is unclear at this point.

As another example, the section at pages 7-8, lines 286-327, discusses Rad53 binding at gene bodies, rather than promoters. This is an interesting finding, but it distracts from the main message of the paper.

Page 5: The authors separate their ChIP-seq experiments into two groups, or sets: the CP set (WT, Rad53 mutant, and mrc1D) and the TP set (ixr1D, swi4D, swi6D and WT). While the conditions in the CP set are the ones used throughout the paper, the motivation behind the TP set in unclear. How were these particular factors selected? And are the "WT" assays the same in the two groups? For the transcription factors, the authors say they are selected "based on the types of genes that bind Rad53", but no other details are provided.

How was differential gene expression performed? (Figure 5)

Line 343: what does "enrichment score" refer to here? For this analysis, Fisher's exact test seems to be the most appropriate test for assessing enrichment of the Swi4/Swi6 targets among the differentially expressed genes.

For scatterplot analyses, e.g. Figure 8, there is no statistical analysis of the observed correlations. What are the R2 values and the corresponding p-values for these correlations?

As mentioned above, the choice of transcription factor null mutants for the Rad53 ChIP-seq experiments is not clearly motivated. SBF (Swi6+Swi4) is definitely a relevant choice, but would this be the top choice in an unbiased analysis of transcription factor proteins? Data on transcription factor gene targets is available for many factors in yeast. An unbiased analysis across all transcription factors could be performed to determine which factor shows the most significant (Fisher's test?) overlap with gene promoters bound by Rad53. In such an analysis, would SBF be the top candidate? If so, this kind of analysis would significantly strengthen the manuscript. Similarly, how was Ixr1 chosen? Was it meant as a control?

*Reviewer #2:*

In the manuscript entitled "Prevalent and Dynamic Binding of the Cell Cycle Checkpoint Kinase Rad53 to Gene Promoters" the authors explore the possibility that occupancy by the replication machinery influences regulation of genes. To do this, the authors measure the occupancy of Rad53, a key signaling kinase for the DNA replication checkpoint (DRC) that binds to replication origins. The authors measure occupancy after putting cells under stress with hydroxyurea in both wild-type cells and in cells in which the DRC is impaired through a mutation in Rad53 and by deletion of Mrc1. With an intact DRC, early origins of replication are activated and are occupied by Rad53. When the DRC is impaired early origins of replication are also activated and are occupied by Rad53, but late origins of replication are activated and occupied as well, indicating a bypass of the DRC. In a key finding, it is also noted that Rad53 occupies regions that are upstream and downstream of a large number of genes, and that the expression of thousands of genes is altered when the DRC is impaired. A difference in Rad53 occupancy near genes often correlates with changes in the expression of these genes. In particular, it appears that an increase in Rad53 binding close to the transcription start site of genes, both upstream and downstream, is correlated with the reduced expression of these genes. A deeper analysis of SBF target genes related to cell cycle and mating-type switching shows a correlation between Rad53 occupancy and gene expression. This appears to show two roles for Rad53 occupancy: inhibition of transcription at some genes when checkpoints are impaired, but also potentially as a transcriptional activator at others. With these roles, the authors assert that Rad53 coordinates both replication and gene expression under replication stress.

There are two major strengths to this manuscript. The first is that transcription-replication conflicts are increasingly being realized as occurring in eukaryotes, and this manuscript provides an excellent example of when such conflicts might occur. The carefully chosen mutants that bypass DRC even under stress show the aberrant firing of late origins in the proximity of genes whose expression is impaired. This provides a relevant and useful system to study conflicts when replication and transcription are not coordinated. The second major strength of this manuscript is the thorough collection of rich data sets in this system. The correlation of origin activity, Rad53 occupancy, and gene expression in WT and mutant backgrounds before and after administration of HU to induce replication allows probing of correlations between aberrant responses to stress and gene expression. The authors provide useful analyses of these data and demonstrate that bypass-induced binding of Rad53 correlates with reduced gene expression at numerous genes. Thus, their main assertions are supported by the data.

The main overall weaknesses of the paper are that it is difficult to read with little effort spent to make the work accessible outside the field of replication stress, the findings are not always well connected to provide a convincing argument, the figure resolution often makes them unreadable and impossible to interpret, and some statistical analyses are either missing or inappropriate. The weaknesses are detailed below:

There is a widespread use of indefinite quantifiers (some, many, several) to make quantitative inferences. Such as:

Line 200: "most genes" remain constant. How may genes go up, down, and remain constant?

Line 221: "Visual inspection of the ChIP-seq peaks suggested that Rad53 bound to numerous gene promoters and TSSs throughout the genome." How many is numerous? What fraction of TSSs?

Line 305: "Overall, most of the down regulated genes in cluster 1 of this group are situated very close to active origins". Reporting numbers make this more convincing.

Line 316: "More down regulated genes are found when the nearby origins are active." Reporting numbers make this more convincing.

Virtually all of the SBF target analysis.

Without quantification and statistics these amount to anecdotal observations. A quantitative inference requires the exact counting of groups followed by a statistical test.

Comment 1: The introduction is confusing. Line 47: After describing DRC and DDC along with four different kinases, the final sentence states that "the signaling" promotes widespread gene expression changes. All signaling? Rad53 specifically? Line 50 says that Mac1 and Rad53 are essential for cell viability, but Line 52 states that kinase null mutants (presumably either kinase?) are extremely sick. Which is it, sick or dead (essential)? The next sentence states that under bypass conditions (not defined) that sml1- and rad53- cells exhibit a "more severe defect" than mec1- cells, and that implies a role for Rad53 beyond DRC. This is either contradictory to Rad53 being essential, or too much background is left out to understand this pathway under different conditions. In addition, the authors already state that Rad53 has a role in DDC, which is beyond its function in DRC. Later the authors do not do a good job connecting RNR to "upregulating dNTP pools". This introduction may be clear to those familiar with Rad53 already, but not for others including those interested in transcription regulation or other aspects of signaling.

As a smaller point, the authors claim in the last paragraph that Rad53 is localized to 20% of promoters suggesting a "global" role in coordinating the response. 20% is not global – it may indicate a multifaceted role in response but global is a very high bar.

Comment 2: Most of the intro talks about the central role of Mec1 in sensing stress – yet the authors chose to first explore mrc1- cells and their stress response. What's the rationale for choosing that and not mec1-?

Comment 3, Line 84. The title doesn't help the reader understand the result. Having more heading that emphasized the results might help the reader more.

Comment 4: Line 101. The accounting for early, late, and inactive origins is confusing in the text but explained better in the methods. In this accounting early is defined as active in WT and late is defined as active in the mutants but not wild type. However, in Figure 2 origins are ranked according to replication timing define elsewhere (Yabuki 2002). Why the difference in the categorization? Do the classifications match perfectly? If so, why not use the Yabuki classification for both and avoid confusion?

Comment 5, Line 105. The assertion is that rad53_K227A favors late origins over early. Figure 1b shows a scatter plot with signal as the y-axis. It appears that a statistical test has been done to presumably show a significant difference between E (early) and L (late). What test was done (not described in body or figure)? Is the test for the difference in signal? If for signal, it could be that there are strong late ORCs, but more early ORCs – which would be a different test. My reading of "favor" late ORCs would be that there is a greater number of late than early, and that signal is how active the favored ORCs are.

Comment 6, line 138. The meaning of "the status of replisomes" is not defined and potentially broad. Something more specific would be helpful.

Comment 7, Line 146. The evidence of "slower progression" of Cdc45 away from the origin in mrc1- mutants is not clear to me. Could the authors describe what they are seeing to help the reader? The heat maps in Figure 2 Supplement 1 are duplicated in Figure 2 b,d, and f. This is not necessary and could be (was) confusing. Further, the Replication TIme scales in Figures 2b,d, and f seem to overlap between early and late – which is odd.

Comment 8, Line 163 "In contrast, the Mrc1 is not strictly required to induce or maintain -H2A." is a conclusion that is drawn without describing the result. The result appears to be that gammaH2A deposition is strong in the mutant and persists even at late time points.

Comment 9: The speculation in paragraph lines 166-169 would fit better in the discussion.

Comment 10, line 173. "dispersed in late times" – there is only one late time, HU90.

Comment 11, line 183. The claim that Rad53 signal increases from HU45 to HU90 is not 100% clear from the plots, and requires some type of test. Diffbind is great for this. Also – "A similar pattern occurs at the RNR3 promoter" I don't see this labeled in Figure 3.

Comment 12, line 190. How is "upstream" of the promoter defined? 1kb? 10kb? I don't see it in the results or methods.

Comment 13, paragraph 190-200. This is a discussion of Rad53 binding near TSSs. The fractions reported are the fraction of Rad53 peaks near genes. This is interesting, but the abstract reports that Rad53 "coordinates…genome-wide transcription" and binds "20%" of promoters. The number of promoters occupied by Rad53 is not reported here and should be accounted for to make these claims.

Comment 14: Paragraph lines 202-210. I found this paragraph confusing. First, the description of residual analysis in the methods was confusing. Summing in 25bp windows over 500bp upstream of TSS seems an odd way to calculate enrichment within a region, corrected for counts. R packages such as DiffBind do this quite rigorously, handle replicates well, and report differentially enriched regions with p-values. Also, reporting the difference in enrichment in regions between conditions is confusing – why not just enriched or depleted occupancy? I also could not quite follow what was meant by "435 genes identified in both". Please clarify, as much of the manuscript relies on this classification.

Comment 15. Figure 3d is not described in the legend, nor how it was calculated in the methods. This could be a useful plot but needs to be described.

Comment 16. The clustering conclusions in paragraph lines 250-256 draw distinctions that are not evident. The paragraph states that there is little difference in samples in G1, yet rad53-mutant and mrc1- samples cluster together. The paragraph also states that different from G1, rad53-mutant, and mrc1- samples cluster together in HU45 and HU90. The repeats of each mutant cluster together, but are not even always in the same clade (HU90). This interpretation is a stretch.

Comment 17. Lines 267-273. I had to read up on co-expression analysis and clustering. It is not clear to me why co-expression analysis is more useful than simple k-means clustering to identify sets of similarly differentially expressed genes. Perhaps a few words on why this method was chosen.

Comment 18. Line 321: "Furthermore, the bias toward the down regulation is even stronger (>80%) when the nearby origin is in a head-on orientation towards the gene." What is the percent when co-directional? Is the difference significant? It looks close enough that discerning a difference might require a statistical test (Fisher's Exact).

Comment 19. I had a hard time following the analysis of SBF and MBF target genes (page 8). First, an "Enrichment Score" is reported for the inclusion of SBF genes in the DEG set. I don't know what an enrichment score is. A more useful score is whether the number of potential SBF target genes is more likely to be differentially regulated than another random transcriptional complex. This can be calculated using Fisher's exact test. Otherwise, reporting that 36 out of 81 is anecdotal (unless the enrichment score is explained better). Second, Figures 8a bottom were impossible to read, and it was not clear what conclusions were to be drawn from them. Lastly, in paragraph lines 367-382, an analysis of the enrichment near genes in SBF mutants was performed. It is asserted that there is a "significant" difference in the enrichment of SBF targets among Top DBs – how was significance calculated? Then, in a series of pictures (Figure 8b insets) that the position of some enrichment is collapsed in mutants. Is this just visual? Calculating the Z-scores for these points and showing that the Z-scores reduce significantly would be a straightforward way to determine that the occupancy collapses toward the average.

Comment 20, lines 439-440. "Our data is consistent with the possibility that the Rad53 kinase contributes to the transcription regulation as a structural component" I don't know what this means. The manuscript asserts that the activity of Rad53 matters – so what is meant by the assertion that it is a "structural component"?

Comment 21. For rigor and reproducibility, the processing code should be included, preferably with raw data, in a manner that can be validated by the reviewer with one or few commands.

*Reviewer #3:*

In this study the authors find that unexpectedly RAD53 is bound to promotor regions of 20% of all yeast genes. The authors use genome wide ChIP-seq and correlative analysis to gene expression. Because of its known function during replication stress, experiments were designed with replication stress perturbation. However, the authors found RAD53 is bound to the promotor regions in G1 and without stress, and independent of canonical checkpoint signaling, making an analysis of experiments with replication stress somewhat illogical in the context of the specific discovery. Moreover, it introduces multiple additional variables directly affecting gene expression. This challenges the strength of interpretations of the correlations made.

Generally the correlations between RAD53 promotor biding and gene expression (some genes are some not) appear to be low. It is also unclear from the manuscript what matrix the authors use to claim significance of a given correlation.

Moreover, alternative interpretations are not considered. The authors solely investigate RAD53 promotor binding for its correlative effects on gene expression. While the finding that RAD53 is bound to promotor sites in the absence of stress is very interesting and promises for a potential greater implication of repair proteins in the conservation of transcription, promotor site binding of repair proteins have been previously reported Specifically, NEIL1 is preloaded at promotor sites of evolutionary critical genes and shown to promote rapid and efficient repair of promotor sites for conversation of critical genes. Such alternatives interpretation are not considered.

The authors may want to consider discussing (Nucleic Acids Research, Volume 49, Issue 1, 11 January 2021, Pages 221-243,) in the context of their discovery.

The authors use a kinase dead RAD53 mutant and a checkpoint dead mutant strain to suggest that RAD53 promotor binding is independent of a stress response. Recruitment to chromatin has been shown to involve acetylation rather than phosphorylation, as seen for Neil1, thus could be considered as an alternative to phosphorylation as stress signals, and alternative RAD53 mutant strains could be considered.

---

## [Author Response]

[Editors’ note: the authors resubmitted a revised version of the paper for consideration. What follows is the authors’ response to the first round of review.]

Comments to the Authors:We are sorry to say that, after consultation with the reviewers, we have decided that this work will not be considered further for publication by eLife.After discussing the manuscript, the reviewers and the reviewing editor concluded that while the paper reports a series of interesting observations related to Rad53 binding to gene promoters independent of checkpoint signaling, the reviewers found the main message of the paper to be unclear. The paper starts with an attempt to look at RAD53 binding to replicons, but finds binding of RAD53 to promoter sites in G1 and in the absence of replication stress. Despite this intriguing observation, the results are analyzed with respect to stressed replication. As a result, the paper does not address reasons or consequences of the observed phenotype other than transcription/replication collisions.From a technical point of view, all reviewers were concerned with the lack of statistics to draw inferences regarding the central question: Does Rad53 coordinate both replication and transcription in response to stress, and do the mutants disrupt this coordination. While it is often useful, to note a trend that is worth exploring, where numbers are available the validity of these inferences must be drawn to strengthen the conclusions.

Despite criticisms on the previous version of manuscript (previous manuscript) about the clarity of background information and about lacking quantitative inferences and statistics (all have now been addressed in the revised manuscript), it is apparent that reviewers found many aspects of this study valuable with interesting observations and rich datasets. Furthermore, reviewer 2 pointed out that we “provide useful analyses of these data and demonstrate that bypass-induced binding of Rad53 correlates with reduced gene expression at numerous genes. Thus, their main assertions are supported by the data..” And reviewer 1 pointed out that our analysis alleviate the concern [related to the "expression bias" previously reported in ChIP-seq datasets, where false peaks were observed at some highly-expressed genes…]

We have made several changes in the revised manuscript (revised manuscript) and believe that this work merits consideration for publication by *eLife*.

Changes we have made to the manuscript and responses to general points

1) The previous manuscript included ample amount of data and analysis around the replication aspect of the Rad53 functions as these results constituted as logical connection to the prior knowledge of this kinase. Nevertheless, upon further consideration, we realized this part of work, while interesting, has become somewhat a distraction from the unexpected new finding that Rad53 is also functioning at gene promoters. Thus, after careful consideration, we decided to reorganize the presentation and removed panels related to DNA replicaiton from the main figures, mainly Figures 1 and 2 in the previous manuscript. The revised manuscript started directly from the finding of Rad53 localizing to gene promoters in asynchronous cell populations. We believe such presentation would direct the focus more appropriately and make the main message clear. Importantly, the revised manuscript also made quantitative inferences and relevant statistics more explicitly, as recommended by reviewers, to strengthen the conclusions.

2) The reviewers found the previous manuscript hard to read, particularly for readers without specific background knowledge. We acknowledged, indeed, that many criticisms of the previous manuscript arose because of misunderstanding of our presentation. The revised manuscript explained the background more carefully and thus improved the readability for more general audience.

3) Reviewers raised questions about some of less-conventional analysis choices we used in this study. In particular, the use of residual analysis instead of a common tool DiffBind for identification of to identify differential binding of Rad53 to gene promoters. Below, we provide further explanation and evidence that our methods are consistent with, and more appropriate than the conventional one for analysis of our datasets.

Briefly, residual analysis extracted the top 1000 deviated genes as top differential binding (DB) genes from aggregated read coverage while DiffBind computed significant q-values from duplicated signal comparison. We compared the DB genes identified from the two methods using G1 and HU45 datasets in experiment CP and found that the DB genes showed reasonable correlation (Spearman r = 0.7278) and p-value < 2.2e-16 for overlap (both 608, DiffBind only 2254, residual only 392, others 3873). We considered reporting the analysis using DiffBind as suggested by the reviewers, but there are several issues with applying DiffBind for these data sets.

i) Residual analysis controls for input whereas DiffBind does not.

ii) More than 40 % of all genes ended up as DB genes from WT G1 to HU45 in the CP experiment in DiffBind. This likely reflected the increased Rad53 protein level in HU45 thus this method is not ideal to determine which gene promoters are preferentially regulated.

iii) We further checked the reproducibility of DB genes by applying DiffBind to datasets from two independent experiments CP and TF. There were only 81 reproducible DB genes. In contrast, there were 435 reproducible DB genes when the residual analyses were applied to the same datasets.

Thus, we decided to use residual analysis to identify DB genes for further analysis.

Reviewer #1:Strength: Upon first reading of the abstract, there was some concern that the observed Rad53 ChIP-seq at promoters could be related to the "expression bias" previously reported in ChIP-seq data sets, where false peaks were observed at some highly-expressed genes, even for exogeneous non-DNA binding proteins. The authors' analyses, especially lines 229-246, alleviate this concern.

We maintained these analyses that the reviewer deemed a strength in the revised manuscript.

Oftentimes throughout the manuscript the authors mention trends in the data, but without providing actual numbers. E.g. on line 180: "we noticed many Rad53 peaks […] and many of these localized […]"; providing numbers would be helpful for readers to understand the scale of the observed trends. The authors refer to specific figures, but those figures do not contain any numbers either.As another example, at line 198, the authors say "Additional genes show increased Rad53 binding […], but at other promoters Rad53 binding decreases during the same time course […]. However, at most genes Rad53 remains constant." Numbers would be helpful here. Also, how were the increases and decreases called? What statistical tests and cutoffs were used to make these calls?Same questions for page 6, lines 216-217. And also line 221, "numerous gene promoters" – could this be expressed as a fraction of all gene promoters in yeast?

All issues raised in this paragraph are addressed in the context of revised manuscript. Specific points from this paragraph are addressed as two separate points.

1)Concerning points raised for line 180 and line 221 in the previous manuscript: Quantitative inferences for the Rad53 binding to gene promoters are presented in the first section “Rad53 is recruited to genomic loci other than replication origins in proliferating yeast cells”.

There we described analysis that established the claim that Rad53 binds to more than 20 % of all gene promoters in proliferating cells (new manuscript lines 89-112). Furthermore, analysis of ChIP-seq peaks from a synchronized population showed that ~90% of the Rad53 ChIP-seq peaks were found at gene promoters at all stages investigated (Figure 2b).

2) Concerning points raised in line 198 and 216-217 of the previous manuscript:

These are addressed in the Results section “Identification of genes with differential binding (DB genes) of Rad53 at promoters”. There we used residual analysis to identify genes with differential binding of Rad53 at their promoters. (see “Changes we have made to the manuscript and responses to general points” section above for rationale to use residual analysis). Text in 216-217 of the previous manuscript is related to previous figure 3a. The exact figure has been removed as our main focus is on how specific transcription factors, Swi4 and Swi6, impact recruitment of Rad53 to promoters. Nevertheless, the normalized coverage tracks for WT samples in different stages are presented in the revised Figure 2c and the related text in Results section “Binding of Rad53 to upstream TSS changes with the cell cycle stages”. The presentation in revised Figure 2c merely pointed out that, even by visual inspection, signals at promoters of genes such as *RNR1*, *PCL1* and *TOS6* appeared varied depending on the cell cycle stage (G1, HU45 and HU90). This observation was further confirmed by the result of residual analysis described in the later subsection “Identification of genes with differential binding (DB genes) of Rad53 at promoters”. That is, these three genes were among the top ranking deviations in the residual analysis of WT samples in both of the two independent datasets.

The authors used "residual analysis in WT" to identify "the top differentially binding (DB) genes" in each set. Why residual analysis and not an established tool for differential binding, such as DEseq? And how were the different experiments in each set combined? For example, the TP set has 3 transcription factor null mutants and one WT condition. Comparing each mutant to the WT will gives three sets of differentially bound gene promoters; how are these combined? Is the direction of the change always the same across the 3 mutants? If not, what is reported in Figure 4b? What exactly is the y-axis in this figure panel?The authors continue to say "Many of these genes encode proteins involved in cell cycle progression and cell growth". How was this determined? Was a statistical analysis (like GSEA) performed?

Issues raised in this paragraph will be address in three parts.

1) Regarding the reviewer's question “Why residual analysis and not an established tool for differential binding, such as DEseq?”, we believe the reviewer means DiffBind tool instead of DEseq, which is designed to apply a DE-seq2 strategy for ChIP-seq data. To validate the applicability of DiffBind for the samples with varied DNA input, we detected the DB genes in WT from two independent experiments CP and TF using DiffBind. As discussed above, a problem of reproducibility was found for DiffBind results presumably because DiffBind does not control for the input and did not take into account the general trends. See above “Changes we have made to the manuscript and responses to general points” section above for rationale to use residual analysis.

2) We regret the insufficient explanation that led to the questioning by the reviewer “how were the different experiments in each set combined?”. We have modified the description in the revised manuscript and indicated more clearly the datasets used in the analysis in the Results section “Identification of genes with differential binding (DB genes) of Rad53 at promoters”. In the case of our residual analysis for identifying genes with differential binding of Rad53 at promoters, only datasets of WT samples were used. We did not combine different experiments. Rather, each experiment has its own replicates and was analyzed separately and only data of stage G1 and HU45 from the same experiments were used for comparison with each other. The result is shown in Figure 3 of the revised manuscript. The previous 4b (now revised Figure 3b) presented the 435 genes that were within the top 1000 ranking in both of the two independent analyses for datasets in experiment TF and datasets in experiment named CP. In response to reviewer’s specific point, the y-axis numbers in the corresponding figure panel are residuals from analysis of experiment TF and this is now stated in the figure legend in revised manuscript.

3)Among the 435 genes identified, there are genes that have well-documented functions in cell cycle, cell growth, etc. Thus, we merely point this out in the text. The statement is based on prior knowledge rather than the implicating bioinformatical analysis. We believe, in addition to hints from bioinformatics, it is equally justified to use the knowledge from the literatures for direction. After all, these are very well characterized genes.

The relevance of the comparison between Rad53 ChIP-seq and Swi6 ChIP-seq is not entire clear. Did the authors mean to illustrate that although Rad53 binds to promoters, its pattern of binding is different from that of transcription factors? I.e. Rad53 binds a large set of promoters, but the range of binding signals is generally narrow, while transcription factors tend to bind strongly but to fewer promoters? If this was the intended purpose of the analysis (presented at lines 221-227), why was Swi6 selected? Would other cell-cycle transcription factors also be relevant for this type of analysis?

The related data (in revised manuscript Figure 2d) is to show the low specificity of Rad53 binding sites compared with binding sites of Swi6. In other words, Rad53 has a higher coverage for many promoters while Swi6 shows substantially high coverage only for a limited number of promoters, as the reviewer pointed out. Thus, it is likely Rad53 would effect a wider range of genes than the Swi6 regulatory network. Swi6 was chosen for this analysis because in the work presented later in the manuscript, where we found that deletion of Swi6 affected Rad53 recruitment to promoters in a small subset of genes. Furthermore, Swi6 would affect two major cell-cycle transcription factors SBF and MBF. The comparison between Rad53 and Swi6 biding to gene promoters emphasizes the observation that Rad53 binds to many more genes than conventional transcription factors.

The comparisons between Rad53 binding and gene expression (line 248-284) show a clear positive correlation, especially in certain gene subsets/clusters. Further analyses, focused on targets of SBF, MBF, Msn4 and Ste12, confirm these correlations on subsets of genes targeted by these transcription factors. But it is unclear, to this reviewer, whether the authors conclusion that "regulation by SBF appears to be responsible for the correlation between increased Rad53 binding at the promoter and up-regulation of these target genes" is truly supported by the data. The correlations and overlaps are compelling, this is true, but are these significant and do they point to causality?

The issue raised here might be semantic. We agree with the reviewer that the correlation and overlaps do not demonstrate causality and we did not intend to claim such. We have now modified the statement to "regulation by SBF may be responsible for the correlation between increased Rad53 binding at the promoter and up-regulation of these target genes" to simply point out the SBF might have a role in increased Rad53 binding, in addition to its role in up-regulation of SBF targets. And this led us to the next investigation presented in Result section “SBF is a key factor for recruitment of Rad53 to the promoters of its target genes under replication stress condition”. There we presented the experiments to determine the contribution of various transcription regulators in recruitment of Rad53. And, indeed, mutation in SBF did affect Rad53 binding to the gene promoters of SBF targets (revised Figure 8).

In Figure 8b, the authors highlight a few SBF targets that show "significant deviation" from the global trend. (How what the significance assessed?) But these are just the SBF targets in the "Top DB" set, i.e. in differentially bound gene promoters. Since this is how they were selected, shouldn't we expect them to deviate from the global trend? It is not clear how this finding shows that "Rad53 signal changes at these genes depends on SBF". This section (lines 364-283) continues with some examples of specific genes and their Rad53 promoter binding chances. But it is not clear what the general conclusion is according to these data.

We modify the phrase "significant deviation" to “clear deviation” as it merely means that these SBF targets are in the 435 TopDB identified in residual analysis. And, as the reviewer pointed out, “since this is how they were selected”, we indeed expected “them to deviate from the global trend“ in the context of the *WT* datasets. Most importantly, however, is the fact that such deviation is no longer evident for these SBF targets when plotted using the signal data obtained from SBF mutants. This finding demonstrated that Rad53 signal changes at these gene promoters depends on SBF. This is the conclusion according to these data.

Furthermore, we added a new panel (suggested by reviewer #2), showing the Z-score distribution of Rad53 DB residual (G1→HU45) and found substantial deviation of SBF target genes from the rest of the genes in *WT* (revised Figure 8b), while in the *swi4Δ* and *swi6Δ* mutants, the deviation of the SBF targets was closer to other genes.

Note: in the revised manuscript, we use the term “general trend” instead of “global trend” to describe the overall increase of Rad53 ChIP signal in HU samples, likely due to increased Rad53 protein level in cells treated with HU (Figure 2 —figure supplement 1).

One aspect that makes the manuscript difficult to read is that there are many observations reported throughout the paper, but it is not always clear how they relate to the main message of the paper. E.g. line 132: "the replication fork collapse was more severe in the absence of Rad53 kinase compared to the absence of checkpoint signaling in the mrcD mutant". This is an interesting observation, but what is the relevance for the main message of the paper?As another example, in the paragraph at line 60 (Introduction), the authors talk about Rfx1 and Ixr1 transcription factors, and the RNR genes. But the relevance of this is unclear at this point.As another example, the section at pages 7-8, lines 286-327, discusses Rad53 binding at gene bodies, rather than promoters. This is an interesting finding, but it distracts from the main message of the paper.

The issues raised here about relevance of results and text at previous line 132… is addressed in the revised manuscript as we removed some of the data related to the replication function of Rad53. Please see above “Changes we have made to the manuscript and responses to general points”. We now focus on Rad53 binding to TSSs in many genes in the genome, which is a novel and very unexpected finding.

About previous line 60 in Introduction, information on *RNR* genes, Rfx1 repressor and Ixr1 are relevant to this study because they constitute some of the well-known responses in gene expression downstream of the Rad53 kinase. Furthermore, in this study, we also found that Rad53 bound to the *RNR1* gene promoter and the binding was reduced in both *ixr1Δ* and SBF mutants (Figure 8).

About lines 286-327, related to the finding of Rad53 binding at gene bodies, we believe it is relevant as it suggests a role for Rad53 in gene regulation in the context of transcription- replication conflicts.

Page 5: The authors separate their ChIP-seq experiments into two groups, or sets: the CP set (WT, Rad53 mutant, and mrc1D) and the TP set (ixr1D, swi4D, swi6D and WT). While the conditions in the CP set are the ones used throughout the paper, the motivation behind the TP set in unclear. How were these particular factors selected? And are the "WT" assays the same in the two groups? For the transcription factors, the authors say they are selected "based on the types of genes that bind Rad53", but no other details are provided.

We regretted that the description and terms for CP and TP sets we used in the previous manuscript caused confusion. All the ChIP-seq experiments using synchronized cultures were done under the same conditions in this study. CP and TF simply referred to two independent experiments. In each independent experiment, WT samples from G1, HU45 and HU90 were included in parallel to samples from mutants at the same stages. The analysis related to what the reviewer was referring to in Page 5 (of previous manuscript) is the datasets used in residual analysis, in which only the datasets from *WT* samples were used (revised Figure 3). Detail of residual analysis done in this study is provided in the result section “Identification of genes with differential binding (DB genes) of Rad53 at promoters” and further detail in the Materials and methods section “ChIP-seq residual analysis”. Mutant datasets were not used in the analysis presented in this part of the study.

About the choice of transcription mutants in ChIP-seq experiment used for further analyses, we selected *swi4Δ* and *swi6Δ* mutants because we noticed SBF targets in both DEGs and Rad53 DB genes in the conditions analyzed (i.e., G1 vs HU45). See more detail in “Rad53 binding changes coincide with the changes in gene expression for targets of cell cycle regulators SBF and MBF” of the Result section. The *ixr1Δ* mutant was also examined because a previous investigation found that Ixr1 binds to the *RNR1* promoter upon genotoxic stress and mediates Dun1- independent *RNR1* gene regulation that requires Rad53, which exhibited dynamic binding at the *RNR1* gene promoter in this study. These choices are explained in the revised paper, citing appropriate literature.

How was differential gene expression performed? (Figure 5)

Differential gene expression was analyzed in a pairwise comparison of RNA-seq data between genotypes (i.e., specific mutant vs WT) under the same conditions (i.e., stage G1, HU45 or HU90) or between different stages (e.g., G1 vs HU45) in the same genotype. See DEG type manifestation in revised Figure 4b (related to previous Figure 5). Detail of RNA-seq analysis is described in “Computational analysis of sequence data” and “Gene expression analysis” in the Materials and methods section.

Line 343: what does "enrichment score" refer to here? For this analysis, Fisher's exact test seems to be the most appropriate test for assessing enrichment of the Swi4/Swi6 targets among the differentially expressed genes.

We agreed with the point of the reviewer and have now presented Fisher's exact test for the enrichment of the transcription factor targets in the bottom panel of Figure 7a.

For scatterplot analyses, e.g. Figure 8, there is no statistical analysis of the observed correlations. What are the R2 values and the corresponding p-values for these correlations?

We agreed with the point of the reviewer and have now presented Spearman correlation r and the corresponding p-values under the scatterplots in Figure 7a (related to previous Figure 8a).

As mentioned above, the choice of transcription factor null mutants for the Rad53 ChIP-seq experiments is not clearly motivated. SBF (Swi6+Swi4) is definitely a relevant choice, but would this be the top choice in an unbiased analysis of transcription factor proteins? Data on transcription factor gene targets is available for many factors in yeast. An unbiased analysis across all transcription factors could be performed to determine which factor shows the most significant (Fisher's test?) overlap with gene promoters bound by Rad53. In such an analysis, would SBF be the top candidate? If so, this kind of analysis would significantly strengthen the manuscript. Similarly, how was Ixr1 chosen? Was it meant as a control?

The choice of the transcription factor null mutants for the Rad53 ChIP-seq experiments has been addressed in response to an earlier point by the same reviewer. We agreed with the reviewer that the choice was not unbiased but a candidate method based on prior knowledge.

While unbiased analysis across all transcription factors could be valuable, the data on transcription factor gene targets are compilations of various large scale studies and manually curated results and not filtered in the same way. Thus, the comparison between data of different transcription would be difficult. The choice of *ixr1Δ* mutant is addressed above to the same reviewer. In paragraph starting “About the choice of transcription mutants in ChIP-seq experiment….”

Reviewer #2:In the manuscript entitled "Prevalent and Dynamic Binding of the Cell Cycle Checkpoint Kinase Rad53 to Gene Promoters" the authors explore the possibility that occupancy by the replication machinery influences regulation of genes. To do this, the authors measure the occupancy of Rad53, a key signaling kinase for the DNA replication checkpoint (DRC) that binds to replication origins. The authors measure occupancy after putting cells under stress with hydroxyurea in both wild-type cells and in cells in which the DRC is impaired through a mutation in Rad53 and by deletion of Mrc1. With an intact DRC, early origins of replication are activated and are occupied by Rad53. When the DRC is impaired early origins of replication are also activated and are occupied by Rad53, but late origins of replication are activated and occupied as well, indicating a bypass of the DRC. In a key finding, it is also noted that Rad53 occupies regions that are upstream and downstream of a large number of genes, and that the expression of thousands of genes is altered when the DRC is impaired. A difference in Rad53 occupancy near genes often correlates with changes in the expression of these genes. In particular, it appears that an increase in Rad53 binding close to the transcription start site of genes, both upstream and downstream, is correlated with the reduced expression of these genes. A deeper analysis of SBF target genes related to cell cycle and mating-type switching shows a correlation between Rad53 occupancy and gene expression. This appears to show two roles for Rad53 occupancy: inhibition of transcription at some genes when checkpoints are impaired, but also potentially as a transcriptional activator at others. With these roles, the authors assert that Rad53 coordinates both replication and gene expression under replication stress.There are two major strengths to this manuscript. The first is that transcription-replication conflicts are increasingly being realized as occurring in eukaryotes, and this manuscript provides an excellent example of when such conflicts might occur. The carefully chosen mutants that bypass DRC even under stress show the aberrant firing of late origins in the proximity of genes whose expression is impaired. This provides a relevant and useful system to study conflicts when replication and transcription are not coordinated. The second major strength of this manuscript is the thorough collection of rich data sets in this system. The correlation of origin activity, Rad53 occupancy, and gene expression in WT and mutant backgrounds before and after administration of HU to induce replication allows probing of correlations between aberrant responses to stress and gene expression. The authors provide useful analyses of these data and demonstrate that bypass-induced binding of Rad53 correlates with reduced gene expression at numerous genes. Thus, their main assertions are supported by the data.The main overall weaknesses of the paper are that it is difficult to read with little effort spent to make the work accessible outside the field of replication stress, the findings are not always well connected to provide a convincing argument, the figure resolution often makes them unreadable and impossible to interpret, and some statistical analyses are either missing or inappropriate. The weaknesses are detailed below:There is a widespread use of indefinite quantifiers (some, many, several) to make quantitative inferences. Such as:Line 200: "most genes" remain constant. How may genes go up, down, and remain constant?Line 221: "Visual inspection of the ChIP-seq peaks suggested that Rad53 bound to numerous gene promoters and TSSs throughout the genome." How many is numerous? What fraction of TSSs?Line 305: "Overall, most of the down regulated genes in cluster 1 of this group are situated very close to active origins". Reporting numbers make this more convincing.Line 316: "More down regulated genes are found when the nearby origins are active." Reporting numbers make this more convincing.Virtually all of the SBF target analysis.Without quantification and statistics these amount to anecdotal observations. A quantitative inference requires the exact counting of groups followed by a statistical test.

We have addressed the overall weaknesses in the revised manuscript. The revised manuscript explains the background more carefully and thus improves the readability for a more general audience. In addition, we have presented relevant statistics more explicitly in the text and figures. Many thanks to the suggestion of the reviewer. See also above “Changes we have made to the manuscript and responses to general points”.

The numbers that the reviewer deem necessary in these series of examples, when possible, were indeed presented in the text or charts in the figures of the previous manuscript despite the use of the indefinite quantifiers (some, many, several) as the reviewer pointed out and we appreciated this. For example, as cited by the reviewer in the next comment, “…Rad53 is localized to 20% of promoters…” In the revised manuscript, we have made further improvement, in particular the presentation of statistical tests and we have also refrained from using the indefinite quantifiers.

More detailed response to comments of reviewer #2 below.

Comment 1: The introduction is confusing. Line 47: After describing DRC and DDC along with four different kinases, the final sentence states that "the signaling" promotes widespread gene expression changes. All signaling? Rad53 specifically? Line 50 says that Mac1 and Rad53 are essential for cell viability, but Line 52 states that kinase null mutants (presumably either kinase?) are extremely sick. Which is it, sick or dead (essential)? The next sentence states that under bypass conditions (not defined) that sml1- and rad53- cells exhibit a "more severe defect" than mec1- cells, and that implies a role for Rad53 beyond DRC. This is either contradictory to Rad53 being essential, or too much background is left out to understand this pathway under different conditions. In addition, the authors already state that Rad53 has a role in DDC, which is beyond its function in DRC. Later the authors do not do a good job connecting RNR to "upregulating dNTP pools". This introduction may be clear to those familiar with Rad53 already, but not for others including those interested in transcription regulation or other aspects of signaling.As a smaller point, the authors claim in the last paragraph that Rad53 is localized to 20% of promoters suggesting a "global" role in coordinating the response. 20% is not global – it may indicate a multifaceted role in response but global is a very high bar.

We thank the reviewer for pointing out specifically the parts in the Introduction that were confusing for more general audience. In response, we have carefully rewrote the background section and made better explanation on connecting the functional relationship among *RAD53*, *RNR1* and *SML1* genes and their roles in regulating cellular dNTP level.

For the next smaller point, in the revised manuscript, instead of “global”, we used “multifaceted”, which indeed depicts more appropriately how we envision Rad53 would function. Great suggestion from the reviewer.

Comment 2: Most of the intro talks about the central role of Mec1 in sensing stress – yet the authors chose to first explore mrc1- cells and their stress response. What's the rationale for choosing that and not mec1-?

The focus of the study is Rad53. In the Introduction, Mec1 was mentioned as the upstream kinase of Rad53 and not really emphasized beyond this point. In the revised manuscript, we indeed started with a result showing that Mec1 is not required for the promoter binding of Rad53 (Figure 1b). We studied Mrc1 because of its specific function in the DRC branch that mediated the signal in response to HU stress. In the context of the revised manuscript, *mrc1Δ* mutant is a great tool for providing a window through which we could examine the effect of untimely activation of late origins on transcription of nearby genes.

Comment 3, Line 84. The title doesn't help the reader understand the result. Having more heading that emphasized the results might help the reader more.

We agree with the point of the reviewer. In the revised manuscript, we have used heading that emphasized the results for the subsection title.

Comment 4: Line 101. The accounting for early, late, and inactive origins is confusing in the text but explained better in the methods. In this accounting early is defined as active in WT and late is defined as active in the mutants but not wild type. However, in Figure 2 origins are ranked according to replication timing define elsewhere (Yabuki 2002). Why the difference in the categorization? Do the classifications match perfectly? If so, why not use the Yabuki classification for both and avoid confusion?

We define early, late, and inactive origins according to the activity of replication origins from our data on de novo DNA synthesis from origins. But this classification does not give us information of time order in continuum. Yabuki’s study, on the other hand, measured time order of chromosome replication rather than the activation time of each origin, though the replication time and origin activation time would largely correlate with each other. Furthermore, their data were measured under different growth conditions. We used their result as a guide to order the origins on the heatmaps for presentation purpose but did not use their data in specific analyses. Overall, the two classification agree with each other.

Comment 5, Line 105. The assertion is that rad53_K227A favors late origins over early. Figure 1b shows a scatter plot with signal as the y-axis. It appears that a statistical test has been done to presumably show a significant difference between E (early) and L (late). What test was done (not described in body or figure)? Is the test for the difference in signal? If for signal, it could be that there are strong late ORCs, but more early ORCs – which would be a different test. My reading of "favor" late ORCs would be that there is a greater number of late than early, and that signal is how active the favored ORCs are.

Mann-Whitney test has been done to show a significant difference in EdU signal (i.e. nucleotide incorporation) between E (early) and L (late) origins. The interpretation is there is more DNA synthesis from late origins observed than from early origins in this particular mutant up to the time when samples were collected. ORC is the Origin Recognition Complex that binds replication origins in a sequence-specific manner as the first step in DNA replication. But perhaps the reviewer meant origin here. We thank the reviewer for the question and will be more careful with the description when presenting this result in the future as this specific result related to DNA replication is no longer presented in the revised manuscript.

Comment 6, line 138. The meaning of "the status of replisomes" is not defined and potentially broad. Something more specific would be helpful.Comment 7, Line 146. The evidence of "slower progression" of Cdc45 away from the origin in mrc1- mutants is not clear to me. Could the authors describe what they are seeing to help the reader? The heat maps in Figure 2 Supplement 1 are duplicated in Figure 2 b,d, and f. This is not necessary and could be (was) confusing. Further, the Replication TIme scales in Figures 2b,d, and f seem to overlap between early and late – which is odd.Comment 8, Line 163 "In contrast, the Mrc1 is not strictly required to induce or maintain -H2A." is a conclusion that is drawn without describing the result. The result appears to be that gammaH2A deposition is strong in the mutant and persists even at late time points.Comment 9: The speculation in paragraph lines 166-169 would fit better in the discussion.Comment 10, line 173. "dispersed in late times" – there is only one late time, HU90.

Most of the result and discussion on replication is no longer presented in the revised manuscript. But we appreciate the reviewer’s careful reading and helpful suggestion and will use them as guidance for presentation of this result in the future. Furthermore, we would like to clarify the issue the reviewer raised about the presentation of heatmaps around origins. In the previous manuscript, the heatmaps in Figure 2 Supplement 1 and in Figure 2 b, d and f were not duplicated but different presentations of the same dataset. In the previous Figure 2 Supplement 1, the origins were ordered according to the replication timing determine in Yabuki’s study. (This presentation is now in Figure 2 Supplement 3 of the revised manuscript). In the previous Figure 2, we separated origins into two categories, i.e. early and late origins, according to our EdU incorporation data, and then ordered origins in each group according to Yabuki’s timing. As explained in the response to comment 4 (see above), the two classification agree with each other. However, disagreement can be anticipated in situations where passive replication from and adjacent early origins renders chromosomal DNA containing late origins becoming early-replicating regions.

Thus, it is expected to have some overlap in the time scales for early and late origins using Yabuki’s timing order.

Comment 11, line 183. The claim that Rad53 signal increases from HU45 to HU90 is not 100% clear from the plots, and requires some type of test. Diffbind is great for this. Also – "A similar pattern occurs at the RNR3 promoter" I don't see this labeled in Figure 3.

The reviewer is referring to the presentation of coverage tracks showing Rad53 binding signal increased from HU45 to HU90 at *RNR1*, *TOS6*, *PCL1* gene promoters. *RNR3* was also mentioned and its coverage tracks showing similar pattern were presented in a later figure. All the example genes mentioned here are DB genes identified by residual analysis (435 Top DB overlap). These genes also tested significant using DiffBind. However, we decided to use residual analysis instead of the DiffBind as the method to identify DB genes in this study. See “Changes we have made to the manuscript and responses to general points” section above for rationale to use residual analysis.

Labelling issue mentioned here no longer exists in the revised manuscript.

Comment 12, line 190. How is "upstream" of the promoter defined? 1kb? 10kb? I don't see it in the results or methods.

Here, we believe the reviewer was asking about how “upstream” of gene is defined in the pie charts showing distribution of Rad53 peaks in relation to genes. In the revised manuscript, we replaced these charts with distribution of aggregated peak numbers around TSS in Figure 2b. The analysis was computed using ChIPpeakAnno, which is cited in Materials and methods section. The peak annotation is obtained by annotatePeakInBatch function and the distribution of the distance to the transcription start sites (TSS) plotted using merged Rad53 ChIP-seq peaks from all three stages (G1, HU45 and HU90). See also related response to Comment 13.

Comment 13, paragraph 190-200. This is a discussion of Rad53 binding near TSSs. The fractions reported are the fraction of Rad53 peaks near genes. This is interesting, but the abstract reports that Rad53 "coordinates…genome-wide transcription" and binds "20%" of promoters. The number of promoters occupied by Rad53 is not reported here and should be accounted for to make these claims.

This is related to Comment 12, this analysis, as pointed out by the reviewer reported the fraction of Rad53 peaks near genes. But the number of promoters occupied by Rad53 (about 20 %) is a different question and is now analyzed by calculating empirical p-value for the ChIP- seq signal 500 bp upstream of TSS. (see “Rad53 is recruited to genomic loci other than replication origins in proliferating yeast cells” in result section of revised manuscript for detail)

Comment 14: Paragraph lines 202-210. I found this paragraph confusing. First, the description of residual analysis in the methods was confusing. Summing in 25bp windows over 500bp upstream of TSS seems an odd way to calculate enrichment within a region, corrected for counts. R packages such as DiffBind do this quite rigorously, handle replicates well, and report differentially enriched regions with p-values. Also, reporting the difference in enrichment in regions between conditions is confusing – why not just enriched or depleted occupancy? I also could not quite follow what was meant by "435 genes identified in both". Please clarify, as much of the manuscript relies on this classification.

We appreciate the reviewer’s comment about our presentation of analysis to identify genes with differential binding of Rad53 at their promoter. In the revised manuscript, the process of this analysis is presented in the Results section “Identification of genes with differential binding (DB genes) of Rad53 at promoters” and explained in more clarity. To detect differentially binding at gene promoters, we computed the read coverage for each 25 bp window due to the limitation of computational resource and time for deepTools. Note that this window size is a half of their default size 50 bp. We focused on the 500 bp window upstream from each TSS where gene promoters are expected to reside in *S. cerevisiae* and applied residual analysis to compare G1 and HU45 signals using datasets from two independent experiments in order to improve the reproducibility of our DB detection. We identified in each analysis the top 1000 ranking DB genes. Among them, 435 genes were found in both independent experiments (revised Figure 3a and b).

Comment 15. Figure 3d is not described in the legend, nor how it was calculated in the methods. This could be a useful plot but needs to be described.

Reference on the calculation of Gini indices and Lorenz curves is described in “Computational analysis of sequence data” of Materials and methods section. Lorenz curve is now in Figure 2d of the revised manuscript.

Comment 16. The clustering conclusions in paragraph lines 250-256 draw distinctions that are not evident. The paragraph states that there is little difference in samples in G1, yet rad53-mutant and mrc1- samples cluster together. The paragraph also states that different from G1, rad53-mutant, and mrc1- samples cluster together in HU45 and HU90. The repeats of each mutant cluster together, but are not even always in the same clade (HU90). This interpretation is a stretch.

This comment is about the hierarchical clustering of rank data analysis of RNA-seq datasets from cell samples of 4 different genotypes and each with 3 growth stages, now presented at Figure 4a of the revised manuscript. It is evident that G1, HU45 and HU90 samples each form a group, with HU45 and HU90 group together at higher level. Within the HU45 group, it is also clear that *rad53^K227A^* and *mrc1Δ* clustered together. We agree with reviewer’s observation on the clustering within HU90 group and have revised accordingly in the revised manuscript. Nevertheless, it is still evident that *rad9Δ* and WT are closer to each other than to *rad53^K227A^* and *mrc1Δ*.

Comment 17. Lines 267-273. I had to read up on co-expression analysis and clustering. It is not clear to me why co-expression analysis is more useful than simple k-means clustering to identify sets of similarly differentially expressed genes. Perhaps a few words on why this method was chosen.

We chose to use co-expression analysis because it took advantage of accumulated knowledge of sequence databases. This is mentioned in the revised manuscript. Moreover, co- expression analysis can suggest a de novo gene-gene relationship that is not annotated in GO. The k-means clustering is applicable to our datasets although it would not find meaningful clusters due to low variability (only two similar conditions are available, one is *WT-rad9Δ* and another is *rad53^K227A^-mrc1Δ*). The results will be also highly confounded with the stage- dependent difference rather than their functional similarities.

Comment 18. Line 321: "Furthermore, the bias toward the down regulation is even stronger (>80%) when the nearby origin is in a head-on orientation towards the gene." What is the percent when co-directional? Is the difference significant? It looks close enough that discerning a difference might require a statistical test (Fisher's Exact).

We appreciated the suggestion of the reviewer and have included Fisher’s exact test *p*- value in the text where the numbers were described. See “Checkpoint mutants cause down- regulation of gene expression near promiscuously active late origins” in Results section.

Comment 19. I had a hard time following the analysis of SBF and MBF target genes (page 8). First, an "Enrichment Score" is reported for the inclusion of SBF genes in the DEG set. I don't know what an enrichment score is. A more useful score is whether the number of potential SBF target genes is more likely to be differentially regulated than another random transcriptional complex. This can be calculated using Fisher's exact test. Otherwise, reporting that 36 out of 81 is anecdotal (unless the enrichment score is explained better). Second, Figures 8a bottom were impossible to read, and it was not clear what conclusions were to be drawn from them. Lastly, in paragraph lines 367-382, an analysis of the enrichment near genes in SBF mutants was performed. It is asserted that there is a "significant" difference in the enrichment of SBF targets among Top DBs – how was significance calculated? Then, in a series of pictures (Figure 8b insets) that the position of some enrichment is collapsed in mutants. Is this just visual? Calculating the Z-scores for these points and showing that the Z-scores reduce significantly would be a straightforward way to determine that the occupancy collapses toward the average.

We appreciated the reviewer’s points in this comment and have done the following in the revised manuscript:

1) Enrichment of transcription factor targets in the 236 DB/DEGs (defined in “The relationship between Rad53 promoter binding and gene expression” of the Results section) was presented with *p*-value of Fisher’s exact test underneath the scatter plots in Figure 7a, which also show the correlation between binding changes (DB residual) and expression changes (log2 FC) of the same transcription factor targets.

2) After careful consideration, we decided to remove the plots at bottom panel of previous Figure 8a as we believe that they merely provided more data without adding major insight.

3) Concerning the paragraph in lines 367-382 of previous manuscript: The analysis and scatter plots (Figure 8a of revised manuscript) presented in this section were comparing the ChIP-seq signal for all genes between G1 and HU45 for WT and transcription factor mutants. We did not say [“there is a "significant" difference in the enrichment of SBF targets among Top DBs] as stated by the reviewer. (Perhaps the reviewer is referring to the same section addressed above in points (1) and (2) of this comment?) Instead, we pointed out there that SBF targets in TOP DB showed “significant” deviation from the general trend in the WT data. The ChIP-seq signal for SBF targets were shown in colors different from the rest of genes in the scatter plots include ChIP-seq signal for all genes. The presentation is also visual and the lack of deviation is evident in the SBF mutants (*swi6Δ* and *swi4Δ*). In the revised manuscript, we use “substantial” deviation instead of “significant” deviation in this context. We agree that it is a good idea to calculate the Z-scores, as suggested by the reviewer, for the DB residuals and have now included the results in Figure 8b of the revised manuscript. Analysis of Z-score distribution for Rad53 DB residual (G1→HU45) also showed substantial deviation of SBF target genes from the rest of the genes in *WT* (Figure 8b), while in the *swi4Δ* and *swi6Δ* mutants, the deviation of the SBF targets was closer to other genes.

Comment 20, lines 439-440. "Our data is consistent with the possibility that the Rad53 kinase contributes to the transcription regulation as a structural component" I don't know what this means. The manuscript asserts that the activity of Rad53 matters – so what is meant by the assertion that it is a "structural component"?

The relevant text is revised as: “Our data is consistent with the possibility that the Rad53 …. as a structural component by binding directly to promoters or transcriptions factors bound to these promoters…” We think it is reasonable to make such speculation since Rad53 is pleiotropic and the importance of the kinase activity does not rule out a kinase-independent role for Rad53.

Comment 21. For rigor and reproducibility, the processing code should be included, preferably with raw data, in a manner that can be validated by the reviewer with one or few commands.

The scripts used in this study including RNA-seq, ChIP-seq, and co-expression analysis are available at https://github.com/carushi/yeast_coexp_analysis.

Reviewer #3:In this study the authors find that unexpectedly RAD53 is bound to promotor regions of 20% of all yeast genes. The authors use genome wide ChIP-seq and correlative analysis to gene expression. Because of its known function during replication stress, experiments were designed with replication stress perturbation. However, the authors found RAD53 is bound to the promotor regions in G1 and without stress, and independent of canonical checkpoint signaling, making an analysis of experiments with replication stress somewhat illogical in the context of the specific discovery. Moreover, it introduces multiple additional variables directly affecting gene expression. This challenges the strength of interpretations of the correlations made.

We respectively disagree with the reviewers comments that the discovery is illogical. Science is advanced by experimenters making observations that are unexpected. We described the study in which we analyzed the localization of Rad53 in the genome using ChIP-seq in the original context of entry into S phase with a HU-induced DNA damage response. We did not expect that Rad53 bound to so many gene promoters in the DNA damage response and certainly did not expect that Rad53 would be bound to TSSs in G1 phase, but this is how discoveries are made. We did place emphasis in the revised paper on Rad53 binding to genes and particularly to TSS regions of genes rather than the origin binding observations to focus the paper more on Rad53 binding to promoters. We showed that in a number of cases, Rad53 binding is influenced by sequence specific transcription factors. This is not, in our view, illogical. Rather it is a novel discovery.

Generally the correlations between RAD53 promotor biding and gene expression (some genes are some not) appear to be low. It is also unclear from the manuscript what matrix the authors use to claim significance of a given correlation.

The reviewer pointed out that “the correlations between RAD53 promotor biding and gene expression (some genes are some not) appear to be low….”. However, the correlations, while not extremely high, are significant for specific subset of genes, catogorized by either co- expression clusters (Figure 5c) or target of specific transcription factors (Figure 7a). Results of statistical tests are included more clearly in the revised manuscript. Furthermore, we find it even more interesting that other co-expression clusters did not show clear correlations, suggesting the possibility these other subsets might be responding to different signals. In the follow up study we plan to investigate the effect of other type of stresses on Rad53 recruitment, dynamic binding and gene expression.

Moreover, alternative interpretations are not considered. The authors solely investigate RAD53 promotor binding for its correlative effects on gene expression. While the finding that RAD53 is bound to promotor sites in the absence of stress is very interesting and promises for a potential greater implication of repair proteins in the conservation of transcription, promotor site binding of repair proteins have been previously reported Specifically, NEIL1 is preloaded at promotor sites of evolutionary critical genes and shown to promote rapid and efficient repair of promotor sites for conversation of critical genes. Such alternatives interpretation are not considered.The authors may want to consider discussing (Nucleic Acids Research, Volume 49, Issue 1, 11 January 2021, Pages 221-243,) in the context of their discovery.The authors use a kinase dead RAD53 mutant and a checkpoint dead mutant strain to suggest that RAD53 promotor binding is independent of a stress response. Recruitment to chromatin has been shown to involve acetylation rather than phosphorylation, as seen for Neil1, thus could be considered as an alternative to phosphorylation as stress signals, and alternative RAD53 mutant strains could be considered.

We agreed with reviewer #3’s suggestion on alternative explanation of Rad53 recruitment to gene promoters. Thus, we have added discussion on the idea of preemptive recruitment of repair factors to potentially vulnerable chromatin regions and also included the recommended reference in the Discussion section. The reviewer also recommended studying alternative rad53 mutant strains, which we have indeed begun to incorporate in our follow up studies. We hope to report these new studies in a new manuscript.

For clarification, we suggested that recruitment of Rad53 to gene promoter does not require the kinase activity *in cis* and nor checkpoint signaling. However, in no way did we try to rule out the contribution of checkpoint signaling or Rad53 kinase activity on a subset of genes. Rather, we suggested that binding can be modulated by multitude of responses eliciated by the experimental condition, not solely the well-defined DRC checkpoint signaling pathway. And the best example is the *RNR1* gene promoter, to which the Rad53 binding is affected by checkpoint response as well as cell cycle regulation (high-lighted in Figure 8a).